# Parallels in the sequential organization of birdsong and human speech

Tim Sainburg [1,2], Brad Theilman[3], Marvin Thielk [3] & Timothy Q. Gentner[1,3,4,5]

Human speech possesses a rich hierarchical structure that allows for meaning to be altered by words spaced far apart in time. Conversely, the sequential structure of nonhuman communication is thought to follow non-hierarchical Markovian dynamics operating over only short distances. Here, we show that human speech and birdsong share a similar sequential structure indicative of both hierarchical and Markovian organization. We analyze the sequential dynamics of song from multiple songbird species and speech from multiple languages by modeling the information content of signals as a function of the sequential distance between vocal elements. Across short sequence-distances, an exponential decay dominates the information in speech and birdsong, consistent with underlying Markovian processes. At longer sequence-distances, the decay in information follows a power law, consistent with underlying hierarchical processes. Thus, the sequential organization of acoustic elements in two learned vocal communication signals (speech and birdsong) shows functionally equivalent dynamics, governed by similar processes.

[1] Department of Psychology, University of California, UC San Diego, La Jolla, CA 92093, USA. [2] Center for Academic Research & Training in Anthropogeny, UC San Diego, La Jolla, CA 92093, USA. [3] Neurosciences Graduate Program, University of California, UC San Diego, La Jolla, CA 92093, USA. [4] Neurobiology Section, Division of Biological Sciences, UC San Diego, La Jolla, CA 92093, USA. [5] Kavli Institute for Brain and Mind, La Jolla, CA 92093, USA. Correspondence and requests for materials should be addressed to T.Q.G. (email: tgentner@ucsd.edu)

Human language is unique among animal communication systems in its extensive capacity to convey infinite meaning through a finite set of linguistic units and rules[1]. The evolutionary origin of this capacity is not well understood, but it appears closely tied to the rich hierarchical structure of language, which enables words to alter meanings across long distances (i.e., over the span of many intervening words or sentences) and timescales. For example, in the sentence, "Mary, who went to my university, often said that she was an avid birder", the pronoun "she" references "Mary", which occurs nine words earlier. As the separation between words (within or between sentences) increases, the strength of these long-range dependencies decays following a power law[2,3]. The dependencies between words are thought to derive from syntactic hierarchies[4,5], but the hierarchical organization of language encompasses more than word- or phrase-level syntax. Indeed, similar power-law relationships exist for the long-range dependencies between characters in texts[6,7], and are thought to reflect the general hierarchical organization of natural language, where higher levels of abstraction (e.g., semantic meaning, syntax, and words) govern organization in lower-level components (e.g., parts of speech, words, and characters)[2,3,6,7]. Using mutual information (MI) to quantify the strength of the relationship between elements (e.g., words or characters) in a sequence (i.e., the predictability of one element revealed by knowing another element), the power-law decay characteristic of natural languages[3,6–8] has also been observed in other hierarchically organized sequences, such as music[3,9] and DNA codons[3,10]. Language is not, however, strictly hierarchical. The rules that govern the patterning of sounds in words (i.e., phonology) are explained by simpler Markovian processes[11–13], where each sound is dependent on only the sounds that immediately precede it. Rather than following a power law, sequences generated by Markovian processes are characterized by MI that decays exponentially, as the sequential distance between any pair of elements increases[3,14]. How Markovian and hierarchical processes combine to govern the sequential structure of speech over different timescales is not well understood.

In contrast to the complexity of natural languages, nonhuman animal communication is thought to be dictated purely by Markovian dynamics confined to relatively short-distance relationships between vocal elements in a sequence[1,15,16]. Evidence from a variety of sources suggests, however, that other processes may be required to fully explain some nonhuman vocal communication systems[17–26]. For example, non-Markovian long-range relationships across several hundred vocal units (extending over 7.5–16.5 min) have been reported in humpback whale song[24]. Hierarchically organized dynamics, proposed as fundamental to sequential motor behaviors[27], could provide an alternate (or additional) structure for nonhuman vocal communication signals. Evidence supporting this hypothesis remains scarce[1,16]. This study examines how Markovian and hierarchical processes combine to govern the sequential structure of birdsong and speech. Our results indicate that these two learned vocal communication signals are governed by similar underlying processes.

## Results

**Modeling**. To determine whether hierarchical, Markovian, or some combination of these two processes better explain sequential dependencies in vocal communication signals, we measured the sequential dependencies between vocal elements in birdsong and human speech. Birdsong (i.e., the learned vocalizations of Oscine birds) is an attractive system to investigate common characteristics of communication signals because birds are phylogenetically diverse and distant from humans, but their songs are spectrally and temporally complex like speech, with acoustic units (notes, motifs, phrases, and bouts) spanning multiple timescales[28]. A number of complex sequential relationships have been observed in the songs of different species[17–23,29]. Most theories of birdsong sequential organization assume purely short timescale dynamics[16,30–32], however, and rely typically on far smaller corpora than those available for written language. Because nonhuman species with complex vocal repertoires often produce hundreds of different vocal elements that may occur with exceptional rarity[21], fully capturing the long-timescale dynamics in these signals is data intensive.

To compare sequential dynamics in the vocal communication signals of birds and humans, we used large-scale data sets of song from four oscine species whose songs exhibit complex sequential organization (European starlings, Bengalese finches[33], Cassin's vireos[21,34], and California thrashers[22,35]). We compared these with large-scale data sets of phonetically transcribed spontaneous speech from four languages (English[36], German[37], Italian[38], and Japanese[39]). To overcome the sparsity in the availability of large-scale transcribed birdsong data sets, we used a combination of hand-labeled corpora from Bengalese finches, Cassin's vireos, and California thrasher, and algorithmically transcribed data sets from European starlings (see "Methods" section; Fig. 1). The full songbird data set comprises 86 birds totaling 668,617 song syllables recorded in over 195 h of total singing (Supplementary Table 1). The Bengalese finch data were collected from laboratory-reared individuals. The European starling song was collected from wild-caught individuals recorded in a laboratory setting. The Cassin's vireo and California thrasher song were collected in the wild[21,34,35,40]. The diversity of individual vocal elements (syllables; a unit of song surrounded by a pause in singing) for an example bird for each species are shown through UMAP[41] projections in Fig. 1a–d, and sequential organization is shown in Fig. 1e–i. For the human speech data sets, we used the Buckeye data set of spontaneous phonetically transcribed American-English speech[36], the GECO data set of phonetically transcribed spontaneous German speech[37], the AsiCA corpus of ortho-phonetically transcribed spontaneous Italian (Calabrian) speech[38], and the CJS corpus of phonetically transcribed spontaneous Japanese speech[39] totaling 4,379,552 phones from 394 speakers over 150 h of speaking (Supplementary Table 2).

For each data set, we computed MI between pairs of syllables or phones, in birdsong or speech, respectively, as a function of the sequential distance between elements (Eq. 4). For example, in the sequence $A \rightarrow B \rightarrow C \rightarrow D$, where letters denote syllable (or phone) categories, $A$ and $B$ have a sequential distance of 1, while $A$ and $D$ have a distance of 3. In general, MI should decay as sequential distance between elements increases and the strength of their dependency drops, because elements separated by large sequential distances are less dependent (on average) than those separated by small sequential distances. To understand the relationship between MI decay and sequential distance in the context of existing theories, we modeled the long-range information content of sequences generated from three different classes of models: a recursive hierarchical model[3], Markov models of birdsong[31,32], and a model combining hierarchical and Markovian processes by setting Markov-generated sequences as the end states of the hierarchical model (Fig. 2). We then compared three models on their fit with the MI decay: a three-parameter exponential decay model (Eq. 5), a three-parameter power-law decay model (Eq. 6), and a five-parameter model which linearly combined the exponential and power-law decay models (composite model; Eq. (7)). Comparisons of model fits were made using the Akaike information criterion (AICc) and the corresponding relative probabilities of each model[42] (see

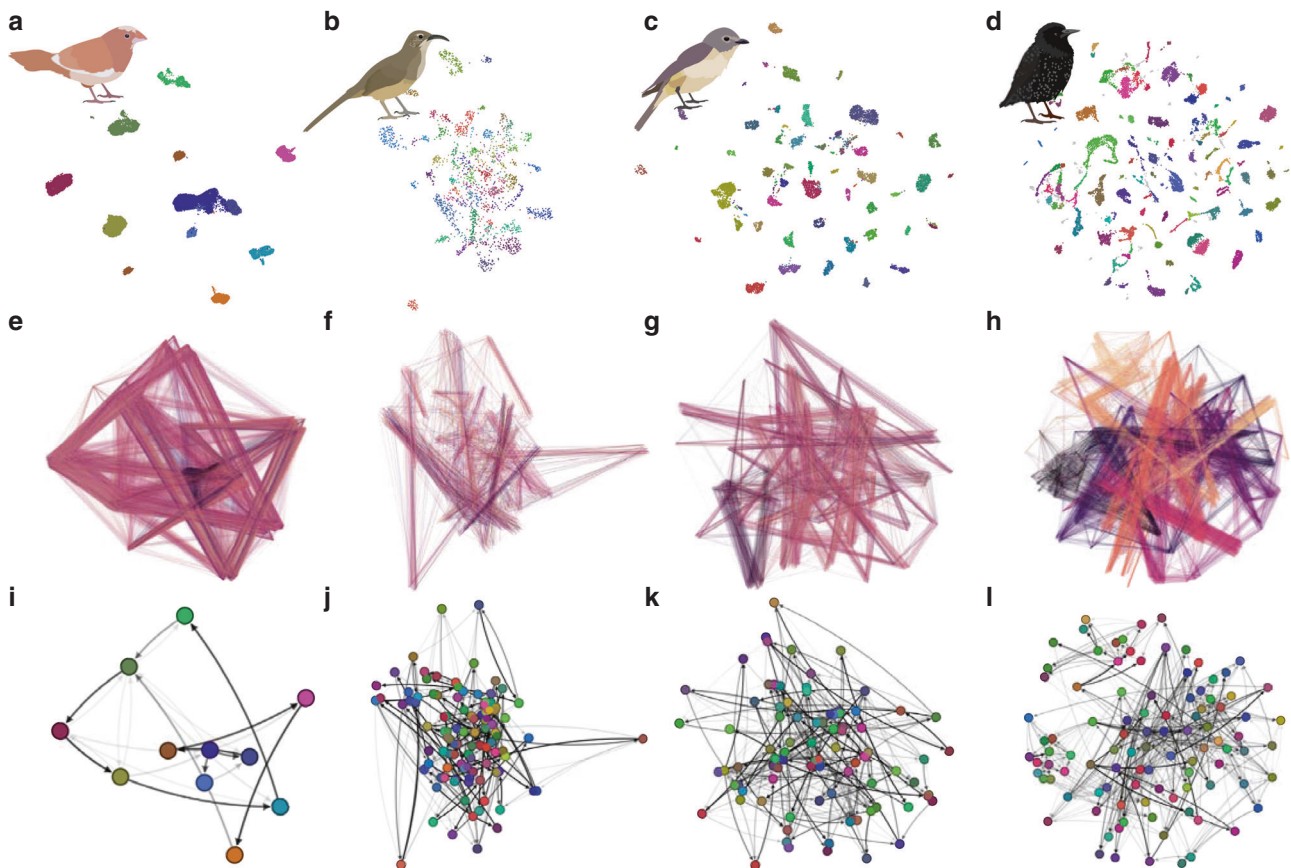

**Fig. 1** Latent and graphical representations of songbird vocalizations. **a–d** show UMAP[41] reduced spectrographic representations of syllables from the songs of single birds projected into two-dimensions. Each point in the scatterplot represents a single syllable, where color is the syllable category. Syllable categories for Bengalese finch (**a**), California thrasher (**b**), and Cassin's vireo (**c**) are hand-labeled. European starlings (**d**) are labeled using a hierarchical density-based clustering algorithm[67]. Each column in the figure corresponds to the same animal. Transitions between syllables (**e–h**) in the same 2D space as **a–d**, where color represents the temporal position of a transition in a song and stronger hues show transitions that occur at the same position; weaker hues indicate syllable transitions that occur in multiple positions. Transitions between syllable categories (**i–l**), where colored circles represents a state or category corresponding to the scatterplots in **a–d**, and lines represent state transitions with opacity increasing in proportion to transition probability. For clarity, low-probability transitions (≤5%) are not shown

"Methods" section) to determine the best-fit model while accounting for the different number of parameters in each model. Consistent with prior work[2,3,8,14], the MI decay of sequences generated by the Markov models is best fit by an exponential decay, while the MI decay of the sequences generated from the hierarchical model is best fit by a power-law decay. For sequences generated by the combined hierarchical and Markovian dynamics, MI decay is best explained by the composite model, that linearly combines exponential and power-law decay (relative probability > 0.999). Because separate aspects of natural language can be explained by Markovian and non-Markovian dynamics, we hypothesized the MI decay observed in human language would be best explained by a pattern of MI decay similar to that observed in the composite model which combines both Markovian and hierarchical processes. Likewise, we hypothesized that Markovian dynamics alone would not provide a full explanation of the MI decay in birdsong.

**Speech**. In all four phonetically transcribed speech data sets, MI decay as a function of inter-phone distance is best fit by a composite model that combines a power-law and exponential decay (Fig. 3, relative probabilities > 0.999, Supplementary Table 3). To understand the relative contributions of the exponential and power-law components more precisely, we measured the curvature of the fit of the log-transformed MI decay (Fig. 3d). The

minimum of the curvature corresponds to a downward elbow in the exponential component of the decay, and the maximum in the curvature corresponds to the point at which the contribution of the power law begins to outweigh that of the exponential. The minimum of the curvature for speech (~3–6 phones for each language or ~0.21–0.31 s) aligns roughly with median word length (3–4 phones) in each language data set (Fig. 3e), while the maximum curvature (~8–13 phones for each language) captures most (~89–99%) of the distribution of word lengths (in phones) in each data set. Thus, the exponential component contributes most strongly at short distances between phones, at the scale of words, while the power law primarily governs longer distances between phones, presumably reflecting relationships between words. The observed exponential decay at inter-word distances agrees with the longstanding consensus that phonological organization is governed by regular (or subregular) grammars with Markovian dynamics[11]. The emphasis of a power-law decay at intra-word distances, likewise, agrees with the prior observations of hierarchical long-range organization in language[12,13].

To more closely examine the language-relevant timescales over which Markovian and hierarchical processes operate in speech, we performed shuffling analyses that isolate the information carried within and between words and utterances in the phone data sets. We defined utterances in English and Japanese as periods of continuous speech broken by pauses in the speech

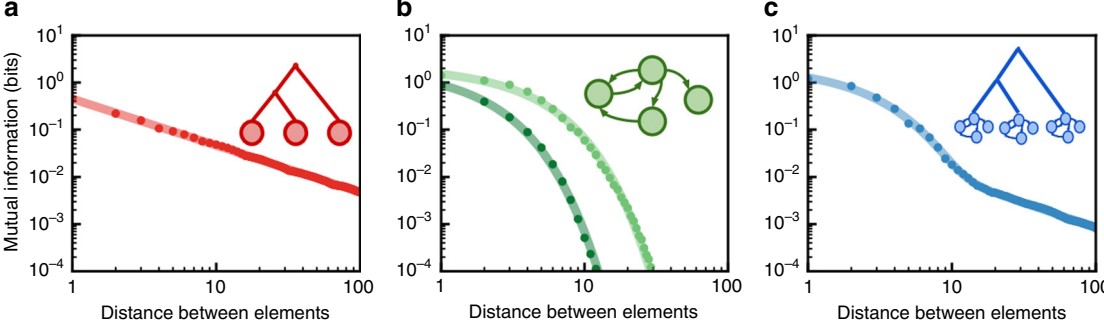

**Fig. 2** MI decay of sequences generated by three classes of models. **a** MI decay of sequences generated by the hierarchically organized model proposed by Lin and Tegmark[3] (red points) is best fit by a power-law decay (red line). **b** MI decay of sequences generated by Markov models of Bengalese finch song from Jin et al.[31] and Katahira et al.[32] (green points) are best fit by an exponential decay model (green lines). **c** MI decay of sequences generated by a composite model (blue points) that combines the hierarchical model (a) and the exponential model (b) is best fit by a composite model (blue line) with both power-law and exponential decays

stream (Supplementary Fig. 1; median utterance length in Japanese: 19 phones, English: 21 phones; the German and Italian data sets were not transcribed by utterance). To isolate within-sequence (word or utterance) information, we shuffled the order of sequences within a transcript, while preserving the natural order of phones within each sequence. Isolating within-word information in this way yields MI decay in all four languages that is best fit by an exponential model (Supplementary Fig. 2a–d). Isolating within-utterance information in the same way yields MI decay best fit by a composite model (Supplementary Fig. 2i, j), much like the unshuffled data (Fig. 3a). Thus, only Markovian dynamics appear to govern phone-to-phone dependencies within words. Using a similar strategy, we also isolated information between phones at longer timescales by shuffling the order of phones within each word or utterance, while preserving the order of words (or utterances). Removing within-word information in this way yields MI decay in English, Italian, and Japanese that is best fit by a composite model and MI decay in German that is best fit by a power-law model (Supplementary Fig. 2e–h). Removing within-utterance information yields MI decay that is best fit by a power-law model (English; Supplementary Fig. 2k) or a composite model (Japanese; Supplementary Fig. 2l). Thus, phone-to-phone dependencies within utterances can be governed by both Markov and/or hierarchical processes. The strength of any Markovian dynamics between phones in different words or utterances weakens as sequence size increase, from words to utterances, eventually disappearing altogether in two of the four languages examined here. The same processes that govern phone-to-phone dependencies also appear to shape dependencies between other levels of organization in speech. We analyzed MI decay in the different speech data sets between words, parts-of-speech, mora, and syllables (depending on transcription availability in each language, see Supplementary Table 2). The MI decay between words was similar to that between phones when within-word order was shuffled. Likewise, the MI decay between parts-of-speech paralleled that between words, and the MI decay between mora and syllables (Supplementary Fig. 3) was similar to that between phones (Fig. 3a). This supports the notion that long-range relationships in language are interrelated at multiple levels of organization[6].

**Birdsong**. As with speech, we analyzed the MI decay of birdsong as a function of inter-element distance (using song syllables rather than phones) for the vocalizations of each of the four songbird species. In all four species, a composite model best fit the MI decay across syllable sequences. (Fig. 4, relative probabilities > 0.999; Supplementary Table 4). The relative contributions of the exponential and power-law components mirrored those observed for phones in speech. That is, the exponential component of the decay is stronger at short syllable-distances, while the power-law component of the decay dominates longer-distance syllable relationships. The transition from exponential to power-law decay (minimum curvature of the fit), was much more variable between songbird species than between languages (Bengalese finch: ~24 syllables or 2.64 s, European starlings ~26 syllables or 19.13 s, Cassin's vireo: ~21 syllables or 48.94 s, California thrasher: ~2 syllables or 0.64 s).

To examine more closely the timescales over which Markovian and hierarchical processes operate in birdsong, we performed shuffling analyses (similar to those performed on speech data sets) that isolate the information carried within and between song bouts. We defined song bouts operationally by inter-syllable pauses based upon the species (see "Methods"). To isolate within-bout information, we shuffled the order of song bouts within a day, while preserving the natural order of syllables within each bout. This yields a syllable-to-syllable MI decay that is best fit by a composite model in each species (Supplementary Fig. 4a–d), similar to that observed in the unshuffled data (Fig. 4). Thus, both Markovian and hierarchical processes operate at within-bout timescales. To confirm this, we also isolated within-bout relationships by computing the MI decay only over syllables pairs that occur within the same bout (as opposed to pairs occurring over an entire day of singing). Similar to the bout shuffling analysis, MI decay in each species was best fit by the composite model (Supplementary Fig. 5). To isolate information between syllables at long timescales, we shuffled the order of syllables within bouts while preserving the order of bouts within a day. Removing within-bout information in this way yields MI decay that is best fit by an exponential decay alone (Supplementary Fig. 4e–h). This contrasts with the results of similar shuffles of phones within words or within utterances in human speech (Supplementary Fig. 2e–i), and suggests that the hierarchical dependencies in birdsong do not extend across song bouts. This may reflect important differences in how hierarchical processes shape the statistics of both communication signals. Alternatively, this may be an uninteresting artifact of the relatively small number of bouts produced by most birds each day (median bouts per day; finch: 117, starling: 13, thrasher: 1, vireo: 3; see "Discussion" section).

To understand how the syntactic organization of song might vary between individual songbirds, even those within the same

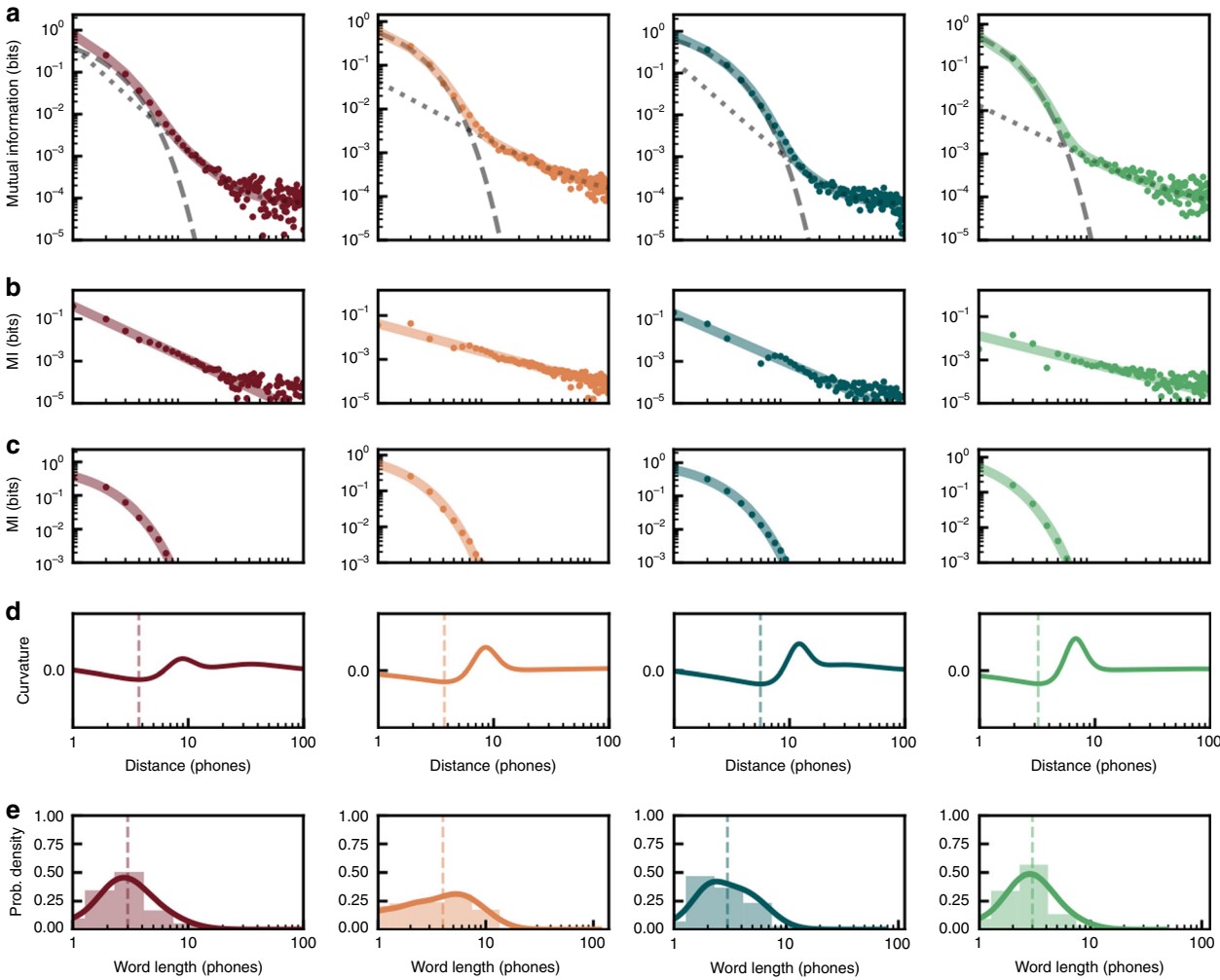

**Fig. 3** Mutual information decay in human speech. **a** MI decay in human speech for four languages (maroon: German, orange: Italian, blue-green: Japanese, green: English) as a function of the sequential distance between phones. MI decay in each language is best fit by a composite model (colored lines) with exponential and power-law decays, shown as a dashed and dotted gray lines, respectively. **b** The MI decay (as in **a**) with the exponential component of the fit model subtracted to show the power-law component of the decay. **c** The same as in **b**, but with the power-law component subtracted to show exponential component of the decay. **d** Curvature of the fitted composite decay model showing the distance (in phones) at which the dominant portion of the decay transitions from exponential to power law. The dashed line is drawn at the minimum curvature for each language (English: 3.37, German: 3.57, Italian: 3.72, Japanese: 5.74) **e** Histograms showing the distribution of word lengths in phones, fit with a smoothed Gaussian kernel (colored line). The dashed vertical line shows the median word length (German: 3, Italian: 4, Japanese: 3, English: 3)

species, we performed our MI analysis on the data from individuals (Supplementary Figs. 6 and 7). One important source of variability is the size of the data set for each individual. In general, the ability of the composite model to explain additional variance in the MI decay over the exponential model alone correlates positively with the total number of syllables in the data set (Supplementary Fig. 7a; Pearson's correlation between (log) data set size and ΔAICc: $r = 0.57$, $p < 0.001$, $n = 66$). That is, for smaller data sets it is relatively more difficult to detect the hierarchical relationships in syllable-to-syllable dependencies. In general, repeating the within-bout and bout-order shuffling analyses on individual songbirds yields results consistent with analyses on the full species data sets (Supplementary Fig. 7b–d). Even in larger data sets containing thousands of syllables, however, there are a number of individual songbirds for whom the composite decay model does not explain any additional variance beyond the exponential model alone (Supplementary Fig. 7). In a subset of the data where it was possible, we also

analyzed MI decay between syllables within a single-day recording session, looking at the longest available recordings in our data set, which were produced by Cassin's vireos and California thrashers and contained over 1000 syllables in some cases (Supplementary Fig. 8). These single-recording sessions show some variability even within individuals, exhibiting decay, that in some cases, appears to be purely dictated by a power law, and in other cases decay is best-fit by the composite model.

## Discussion

Collectively, our results reveal a common structure in both the short- and long-range sequential dependencies between vocal elements in birdsong and speech. For short timescale dependencies, information decay is predominantly exponential, indicating sequential structure that is governed largely by Markovian processes. Throughout vocal sequences, however, and especially for long timescale dependencies, a power law, indicative of

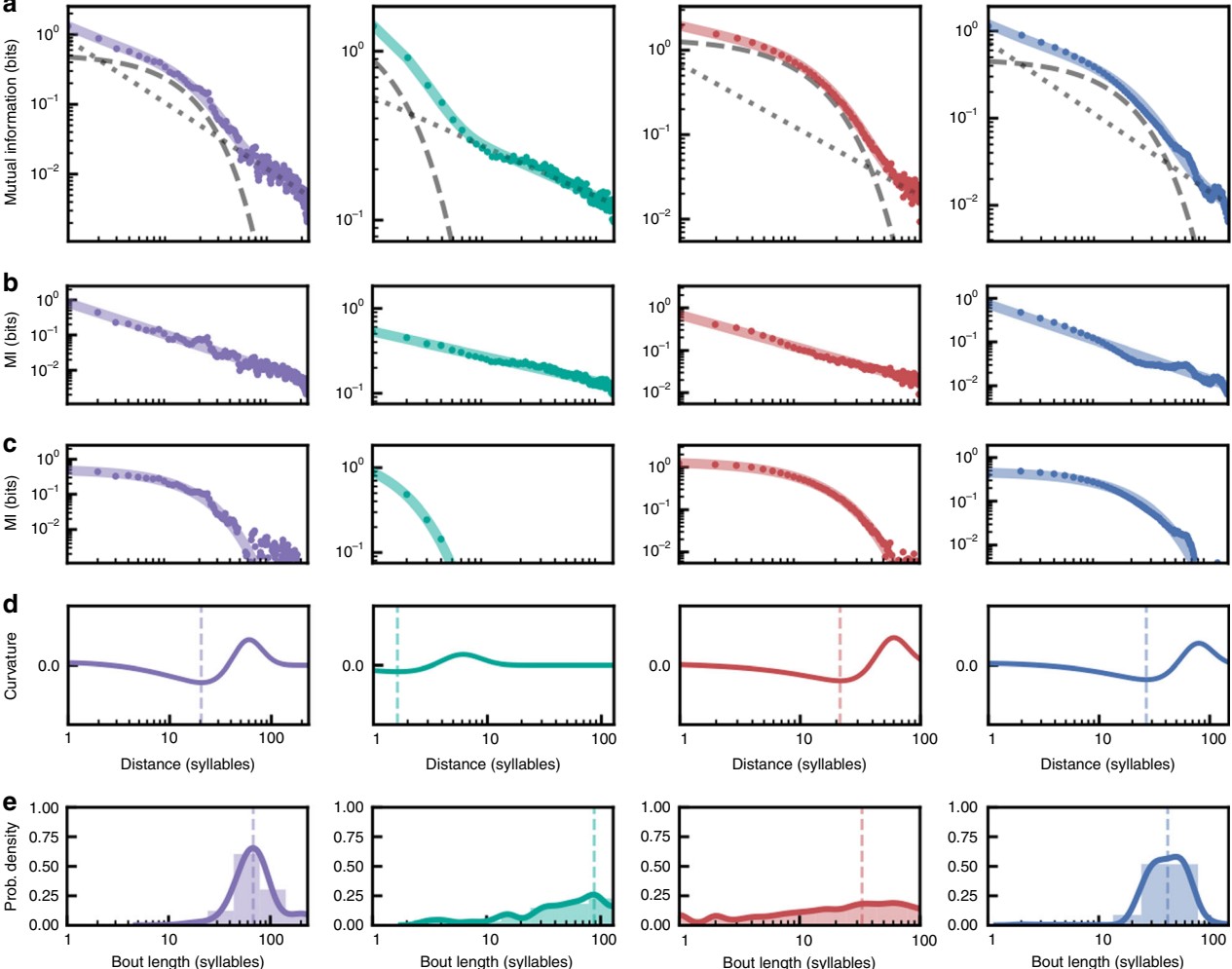

**Fig. 4** Mutual information decay in birdsong. **a** MI decay in song from four songbird species (purple: Bengalese finch, teal: California thrasher, red: Cassin's vireo, blue: European starling) as a function of the sequential distance between syllables. MI decay in each species is best fit by a composite model (colored lines) with exponential and power-law decays, shown as a dashed and dotted gray lines, respectively. **b** The MI decay (as in **a**) with the exponential component of the fit model subtracted to show the power-law component of the decay. **c** The same as in **b**, but with the power-law component subtracted to show exponential component of the decay. **d** Curvature of the fitted composite decay model showing the distance (in syllables) at which the dominant portion of the decay transitions from exponential to power law. The dashed line is drawn at the minimum curvature for each species (Bengalese finch: ~24, California thrasher: ~2, Cassin's vireo: ~21, European starling: ~26) **e** Histograms showing the distribution of bout lengths in syllables, fit with a smoothed Gaussian kernel (colored line). The dashed line shows the median bout length (Bengalese finch: 68, California thrasher: 88, Cassin's vireo: 33, European starling: 42)

non-Markovian hierarchical processes, governs information decay in both birdsong and speech.

These results change our understanding of how speech and birdsong are related. For speech, our observations of non-Markovian processes are not unexpected. For birdsong, they explain a variety of complex sequential dynamics observed in prior studies, including long-range organization[20], music-like structure[19], renewal processes[17,18], and multiple timescales of organization[23,29]. In addition, the dominance of Markovian dynamics at shorter timescales may explain why such models have seemed appealing in past descriptions of birdsong[28,30] and language[43] which have relied on relatively small data sets parsed into short bouts (or smaller segments) where the non-Markovian structure is hard to detect (Supplementary Fig. 7). Because the longer-range dependencies in birdsong and speech cannot be fully explained by Markov models, our observations rule out the notion that either birdsong or speech is fully defined by regular grammars[28]. Instead, we suggest that the organizing principles of birdsong[23], speech[1], and perhaps sequentially patterned behaviors in general[27,44], are better explained by models that incorporate hierarchical organization. The composite structure of the sequential dependencies in these signals helps explain why Hidden Markov Models (HMMs) and Recurrent Neural Networks (RNNs) have been used successfully to model sequential dynamics in speech[3,45–50] and (to a lesser extent) animal communication[29,32,51–57]. HMMs are a class of Markov model which can represent hidden states that underlie observed data, allowing more complex (but still Markovian) sequential dynamics to be captured. HMMs have historically played an important role in speech and language-modeling tasks such as speech synthesis[58] and speech recognition[50], but have recently been overtaken by RNNs[46–49,59], which model long-range dependencies better than the Markovian assumptions underlying HMMs. A similar shift to incorporate RNNs, or other methods to model hierarchical

dynamics, will aid our understanding of at least some nonhuman vocal communication signals.

The structure of dependencies between vocal elements in birdsong and human speech are best described by both hierarchical and Markovian processes, but the relative contributions of these processes show some differences across languages and species. In speech, information between phones within words decays exponentially (Supplementary Fig. 2a–d), while the information within utterances follows a combination of exponential and power-law decay (Supplementary Fig. 2i, j). When this within-word and within-utterance structure is removed (Supplementary Fig. 2), a strong power law still governs dependencies between phones, indicating a hierarchical organization that extends over very long timescales. Like speech, information between syllables within bouts of birdsong are best described by a combination of power-law and exponential decay (Supplementary Figs 5, 7a, b). In contrast to speech, however, we did not observe a significant power-law decay beyond that in the bout-level structure (Supplementary Fig. 7c). The absence of a power law governing syllable dependencies between bouts must be confirmed in future work, as our failure to find it may reflect the fact that we had far fewer bouts per analysis window in the birdsong data sets than we had utterances in the speech data sets. If confirmed, however, it would indicate an upper bound for the hierarchical organization of birdsong. It may also suggest that a clearer delineation exists between the hierarchical and Markovian processes underlying speech than those underlying birdsong. In speech the exponential component of the decay is overtaken by the power-law decay at timescales < 1 s (0.48–0.72 s; Fig. 3a), whereas in birdsong the exponential component remains prominent for, in some cases, over 2 min (2.43–136.82 s; Fig. 4a). In addition to upward pressures that may push the reach of hierarchical processes to shape longer and longer dependencies in speech, there may also be downward pressures that limit the operational range of Markovian dynamics. In any case, words, utterances, and bouts are only a small subset of the many possible levels of transcription in both signals (e.g., note<syllable< motif<phrase<bout<song; phone<syllable<word<-phrase<sentence). Understanding how the component processes that shape sequence statistics are blended and/or separated in different languages and species, and at different levels of organization is a topic for future work. It is also important to note that many individual songbirds produced songs that could be fully captured by Markov processes (Supplementary Fig. 7). In so far as both the Markovian and hierarchical dynamics capture the output of underlying biological production mechanisms, it is tempting to postulate that variation in signal dynamics across individuals and species may reflect the pliability of these underlying mechanisms, and their capacity to serve as a target (in some species) for selective pressure. The songbird species sampled here are only a tiny subset of the many songbirds and nonhuman animals that produce sequentially patterned communication signals, let alone other sequentially organized behaviors and biological processes. It will be important for future work to document variation in hierarchical organization in a phylogenetically controlled manner and in the context of ontogenic experience (i.e., learning). Our sampling of songbird species was based on available large-scale corpora of songbird vocalizations, and most likely does not capture the full diversity of long- and short-range organizational patterns across birdsong and nonhuman communication. The same may hold true for our incomplete sampling of languages.

Our observations provide evidence that the sequential dynamics of human speech and birdsong are governed by both Markovian and hierarchical processes. Importantly, this result does not speak to the presence of any specific formal grammar underlying the structure of birdsong, especially as it relates to the various hierarchical grammars thought to support the phrasal syntax of language. It is possible that the mechanisms governing syntax are distinct from those governing other levels of hierarchical organization. One parsimonious conclusion is that the non-Markovian dynamics seen here are epiphenomena of a class of hierarchical processes used to construct complex signals or behaviors from smaller parts, as have been observed in other organisms including fruit flies[60,61]. These processes might reasonably be co-opted for speech and language production[62]. Regardless of variability in mechanisms, however, the power-law decay in information content between vocal elements is not unique to human language. It can and does occur in other temporally sequenced vocal communication signals including those that lack a well-defined (perhaps any) hierarchical syntactic organization through which meaning is conveyed.

## Methods

**Birdsong data sets**. We analyzed song recordings from four different species: European starling (*Sturnus vulgaris*), Bengalese finch (*Lonchura striata domestica*), Cassin's vireo (*Vireo cassinii*), and California thrasher (*Toxostoma redivivum*). As the four data sets were each hand-segmented or algorithmically segmented by different research groups, the segmentation methodology varies between species. The choice of the acoustic unit used in our analyses are somewhat arbitrary and the choice of the term syllable is used synonymously across all four species in this text, however the units that are referred to here as syllables for the California thrasher and Cassin's vireo are sometimes referred to as phrases in other work[21,22,34,35]. Information about the length and diversity of each syllable repertoire is provided in Extended Data Table 1.

The Bengalese finch data set[33,52] was recorded from sound-isolated individuals and was hand-labeled. The Cassin's vireo[21,34,63] and the California thrasher[35] data sets were acquired from the Bird-DB[40] database of wild recordings, and were recorded from the Sierra Nevada and Santa Monica mountains, respectively. Both data sets are hand-labeled. The European starling song[64] was collected from wild-caught male starlings (sexed by morphological characteristics) 1 year of age or older. Starling song was recorded at either 44.1 or 48 kHz over the course of several days to weeks, at various points throughout the year in sound-isolated chambers. Some European starlings were administered with testosterone before audio recordings to increase singing behavior. The methods for annotating the European starling data set are detailed in the "Corpus annotation for European starlings" section.

Procedures and methods comply with all relevant ethical regulations for animal testing and research and were carried out in accordance with the guidelines of the Institutional Animal Care and Use Committee at the University of California, San Diego.

**Speech corpora**. Phone transcripts were taken from four different data sets: the Buckeye corpus of spontaneous conversational American-English speech[36], the IMS GECO corpus of spontaneous German speech[37], the AsiCA corpus of spontaneous Italian speech of the Calabrian dialect[38] (south Italian), and the CSJ corpus of spontaneous Japanese speech[39].

The American-English speech corpus (Buckeye) consists of conversational speech taken from 40 speakers in Columbus, Ohio. Alongside the recordings, the corpus includes transcripts of the speech and time aligned segmentation into words and phones. Phonetic alignment was performed in two steps: first using HMM automatic alignment, followed by hand adjustment and relabeling to be consistent with the trained human labeler. The Buckeye data set also transcribes pauses, which are used as the basis for boundaries in an utterance in our analyses.

The German speech corpus (GECO) consists of 46 dialogs ~25 min in length each, in which previously unacquainted female subjects are recorded conversing with one another. The GECO corpus is automatically aligned at the phoneme and word level using forced alignment[65] from manually generated orthographic transcriptions. A second algorithmic step is then used to segment the data set into syllables[65].

The Italian speech data (AsiCA) consist of directed, informative, and spontaneous recordings. Only the spontaneous subset of the data set was used for our analysis to remain consistent with the other data sets. The spontaneous subset of the data set consists of 61 transcripts each lasting an average of 35 min. The AsiCA data set is transcribed using a hybrid orthographic/phonetic transcription method where certain phonetic features were noted with International Phonetic Alphabet labels.

The CSJ consists of spontaneous speech from either monologues or conversations which are hand transcribed. We use the core subset of the corpus, both because it is the fully annotated subset of the data set, and because it is similar in size to the other data sets used. The core subset of the corpus contains over 500,000 words annotated for phonemes and several other speech features, and consists primarily of spontaneously spoken monologues. CSJ is also annotated at the level of *mora*, a syllable-like unit consisting of one or more phonemes and

serving as the basis of the 5–7–5 structure of the Haiku[66]. In addition, CSJ is transcribed at the level of Inter-Pausal Units (IPUs) which are periods of continuous speech surrounded by an at-least 200-ms pause. We refer here to IPUs as utterances to remain consistent with the Buckeye data set.

As each of the data sets was transcribed using a different methodology, this disparity between the transcription methods may account for some differences in the observed MI decay. The impact of using different transcription methods are at present unknown. The same disparity is true of the birdsong data sets.

**Corpus annotation for European starlings.** The European starling corpus was annotated using a novel unsupervised segmentation and annotation algorithm being maintained at GitHub.com/timsainb/AVGN. An outline of the algorithm is given here.

Spectrograms of each song bout were created by taking the absolute value of the one-sided short-time Fourier transformation of the band-pass-filtered waveform. The resulting power was normalized from 0 to 1, log-scaled, and thresholded to remove low-amplitude background noise in each spectrogram. The threshold for each spectrogram was set dynamically. Beginning at a base-power threshold, all power in the spectrogram below that threshold was set to zero. We then estimated the periods of silence in the spectrogram as stretches of spectrogram where the sum of the power over all frequency channels at a given time point was equal to zero. If there were no stretches of silence for at least $n$ seconds (described below), the power threshold was increased and the process was repeated until our criteria for minimum length silence was met or the maximum threshold was exceeded. Song bouts for which the maximum threshold was exceeded in our algorithm were excluded as too noisy. This method also filtered out putative bouts that were composed of nonvocal sounds. Thresholded spectrograms were convolved with a Mel-filter, with 32 equally spaced frequency bands between the high and low cutoffs of the Butterworth bandpass filter, then rescaled between 0 and 255.

To segment song bouts into syllables, we computed the spectral envelope of each song spectrogram, as the sum power across the Mel-scaled frequency channels at every time-sample in the spectrogram. We defined syllables operationally as periods of continuous vocalization bracketed by silence. To find syllables, we first marked silences by minima in the spectral envelope and considered the signal between each silence as a putative syllable. We then compared the duration of the putative syllable with an upper bound on the expected syllable length for each species. If the putative syllable was longer than the expected syllable length, it was assumed to be a concatenation of two or more syllables which had not yet been segmented, and the threshold for silence was raised to find the boundary between those syllables. This processes repeated iteratively for each putative syllable until it was either segmented into multiple syllables or a maximum threshold was reached, at which point it was accepted as a long syllable. This dynamic segmentation algorithm is important for capturing certain introductory whistles in the European starling song, which can be several times longer than any other syllable in a bout.

Several hyperparameters were used in the segmentation algorithm. The minimum and maximum expected lengths of a syllable in seconds (ebr_min, ebr_max) were set to 0.25/0.75 s. The minimum number of syllables (min_num_sylls) expected in a bout was set to 20. The maximum threshold for silence (max_thresh), relative to the maximum of the spectral envelope was set to 2%. To threshold out overly noisy song, a minimum length of silence threshold was expected in each bout (min_silence_for_spec), set at 0.5 s. The base spectrogram (log) threshold for power considered to be spectral background noise (spec_thresh) was set at 4.0. This threshold value was set dynamically, where the minimum spectral background noise (spec_thresh_min) was set to be 3.5.

We reshaped the syllable spectrograms to create uniformly sized inputs for the dimensionality reduction algorithm. Syllable time-axes were resized using spline interpolation to match a sampling rate of 32 frames equaling the upper limit of the length of a syllable for each species (e.g., a starling's longest syllables are ~1 s, so all syllables are reshaped to a sampling rate of 32 samples/s). Syllables that were shorter than the set syllabic rate were zero-padded on either side to equal 32-time samples, and syllables that were longer than the upper bound were resized to 32-time samples to fit into the network.

Multiple algorithms exist to transcribe birdsong corpora into discrete elements. Our method is unique in that it does not rely on supervised (experimenter) element labeling, or hand-engineered acoustic features specific to individual species beyond syllable length. The method consists of two steps: (1) project the complex features of each birdsong data set onto a two-dimensional space using the UMAP dimensionality reduction algorithm[41] and (2) apply a clustering algorithm to determine element boundaries[67]. Necessary parameters (e.g. the minimum cluster size) were set based upon visual inspection of the distributions of categories in the two-dimensional latent space. We demonstrate the output of this method in Fig. 1 both on a European starling data set using our automated transcription, and on the Cassin's vireo, California thrasher, and Bengalese finch data sets. The dimensionality reduction procedure was used for the Cassin's vireo, Bengalese finch, and California thrasher data sets, but using hand segmentations rather than algorithmic segmentations of boundaries. The hand labels are also used rather than UMAP labels for these three species.

**Song bouts.** Data sets were either made available, segmented into bouts by the authors of each data set, as in the case of the Bengalese finches, or were segmented

into bouts based upon inter-syllable gaps of >60 s in the case of Cassin's vireo and California thrashers, and 10 s in the case of European starlings. These thresholds were set based upon the distribution of inter-syllable gaps for each species (Supplementary Fig. 9).

**Mutual information estimation.** We calculated MI using distributions of pairs of syllables (or phones) separated by some distance within the vocal sequence. For example, in the sequence "$a \rightarrow b \rightarrow c \rightarrow d \rightarrow e$", where letters denote exemplars of specific syllable or phones categories, the distribution of pairs at a distance of "2" would be $((a, c), (b, d), (c, e))$. We calculate MI between these pairs of elements as:

$$\hat{I}(X, Y) = \hat{S}(X) + \hat{S}(Y) - \hat{S}(X, Y), \tag{1}$$

where $X$ is the distribution of single elements ($a, b, c$) in the example, and $Y$ is the distribution of single elements ($c, d, e$). $\hat{S}(X)$ and $\hat{S}(Y)$ are the marginal entropies of the distributions of $X$ and $Y$, respectively, and $\hat{S}(X, Y)$ is the entropy of the joint distribution of $X$ and $Y$, $((a, c), (b, d), (c, e))$. We employ the Grassberger[68] method for entropy estimation used by Lin and Tegmark[3] which accounts for under-sampling true entropy from finite samples:

$$\hat{S} = \log_2(N) - \frac{1}{N} \sum_{i=1}^{K} N_i \psi(N_i), \tag{2}$$

where $\psi$ is the digamma function, $K$ is the number of categories (e.g. syllables or phones) and $N$ is the total number of elements in each distribution. We account for the lower bound of MI by calculating the MI on the same data set, where the syllable sequence order is shuffled:

$$\hat{I}_{sh}(X, Y) = \hat{S}(X_{sh}) + \hat{S}(Y_{sh}) + \hat{S}(X_{sh}, Y_{sh}), \tag{3}$$

where $X_{sh}$ and $Y_{sh}$ refer to the same distributions as $X$ and $Y$ described above, taken from shuffled sequences. This shuffling consists of a permutation of each individual sequence being used in the analysis, which differs depending on the type of analysis (e.g. a bout of song in the analysis shown in Supplementary Fig. 5 versus an entire day of song in Fig. 4).

Finally, we subtract out the estimated lower bound of the MI from the original MI measure.

$$\text{MI} = \hat{I} - \hat{I}_{sh} \tag{4}$$

**Mutual information decay fitting.** To determine the shape of the MI decay, we fit three decay models to the MI as a function of element distance: an exponential decay model, a power-law decay model, and a composite model of both, termed the composite decay:

$$\text{exponential decay} = a * e^{-x*b} + c \tag{5}$$

$$\text{power} - \text{law decay} = a * x^b + c \tag{6}$$

$$\text{composite decay} = a * e^{-x*b} + c * x^d + f \tag{7}$$

where $x$ represents the inter-element distance between units (e.g., phones or syllables). To fit the model on a logarithmic scale, we computed the residuals between the log of the MI and of the model's estimation of the log of the MI. Because our distances were necessarily sampled linearly as integers, we scaled the residuals during fitting by the log of the distance between elements. This was done to emphasize fitting the decay in log-scale. The models were fit using the lmfit Python package[69].

**Model selection.** We used the Akaike information criterion (AIC) to compare the relative quality of the exponential, composite, and power-law models. AIC takes into account goodness-of-fit and model simplicity, by penalizing larger numbers of parameters in each model (3 for the exponential and power-law models, 5 for the composite model). All comparisons use the AICc[42] estimator, which imposes an additional penalty (beyond the penalty imposed by AIC) to correct for higher-parameter models overfitting on smaller data sets. We choose the best-fit model for the MI decay of each bird's song and the human speech phone data sets using the difference in AICc between models[42]. In the text, we report the relative probability of a given model (in comparison to other models), which is computed directly from the AICc[42] (see Supplementary Information). We report the results using log-transformed data in the main text (Extended Data Tables 3 and 4).

To determine a reasonable range of element-to-element distances for all the birdsong and speech data sets, we analyzed the relative goodness-of-fit (AICc) and proportion of variance explained ($r^2$) for each model on decays over distances ranging from 15 to 1000 phones/syllables apart. The composite model provides the best fit for distances up to at least 1000 phones in each language (Supplementary Fig. 10) and at least the first 100 syllables for all songbird species (Supplementary Fig. 11). To keep analyses consistent across languages and songbird species we report on analyses using distances up to 100 elements (syllables in birdsong and phones in speech). Figures 3 and 4 show a longer range of decay in each language and songbird species, plotted up to element distances where the coefficient of

determination ($r^2$) remained within 99.9% of its value when fit to 100-element distances.

**Curvature of decay fits**. We calculated the curvature for those signals best fit by a composite model in log space (log-distance and log-MI).

$$\kappa = \frac{|y''|}{(1 + y'^2)^{\frac{3}{2}}} \qquad (8)$$

where $y$ is the log-scaled MI. We then found the local minima and the following local maxima of the curvature function, which corresponds to the "knee" of the exponential portion of the decay function, and the transition between a primary contribution on the exponential decay to a primary contribution of the power-law decay.

**Sequence analyses**. Our primary analysis was performed on sequences of syllables that were produced within the same day to allow for both within-bout and between-bout dynamics to be present. To do so, we considered all syllables produced within the same day as a single sequence and computed MI over pairs of syllables that crossed bouts, regardless of the delay in time between the pairs of syllables. In addition to the primary within-day analysis, we performed three controls to observe whether the observed MI decay was due purely to within-bout, or between-bout organization. The first control was to compute the MI between only syllables that occur within the same bout (as defined by a 10 s gap between syllables). Similar to the primary analysis (Fig. 4), the best-fit model for within-bout MI decay is the composite model (Supplementary Figs 7b and 5). To more directly dissociate within-bout and between-bout syllable dependencies in songbirds, we computed the MI decay after removing either within- or between-bout structure. To do this, we shuffled the ordering of bouts within a day while retaining the order of syllables within each bout (Supplementary Fig. 7c), or shuffled the order of syllables within each bout while retaining the ordering of bouts (Supplementary Fig. 7d). Analyses were performed on individual songbirds with at least 150 syllables in their data set (Supplementary Fig. 7), and on the full data set of all birds in a given species. We performed similar shuffling analysis on the speech data sets (Supplementary Fig. 2). For speech, we shuffled the order of phones within words (while preserving word order) to remove within-word information, and shuffled word order (while preserving within-word phone ordering) to remove between-word information. We used a similar shuffling strategy at the utterance level remove within- and between-utterance information. The speech data sets were not broken down into individuals due to limitations in data set size at the individual level, and because language is clearly shared between individuals in each speech data set.

To address the possibility that repeating syllables might account for long-range order, we performed separate analyses on both the original syllable sequences (as produced by the bird) and compressed sequences in which all sequentially repeated syllables were counted as a single syllable. The original and compressed sequences show similar MI decay shapes (Supplementary Fig. 12). We also assessed how our results relate to the timescale of segmentation and discretization of syllables or phones by computing the decay in MI between discretized spectrograms of speech and birdsong at different temporal resolutions (Supplementary Fig. 13) for a subset of the data. Long-range relationships are present throughout both speech and birdsong regardless of segmentation, but the pattern of MI decay does not follow the hypothesized decay models as closely as that observed when the signals are discretized to phones or syllables, supporting the nonarbitrariness of these low-level production units.

**Computational models**. We compared the MI decay of sequences produced by three different artificial grammars: (1) Markov models used to describe the song of two Bengalese finches[31,32], (2) The hierarchical model proposed by Lin and Tegmark[3], and (3) a model composed of both the hierarchical model advocated by Lin and Tegmark and a Markov model. While these models do not capture the full array of possible sequential models and their signatures in MI decay, they well-capture the predictions made based upon the discussed literature[2,3,6,7,14] and provide an illustration of what would be expected given our competing hypotheses. With each model, we generate corpora of sequences, then compute the MI decay of the sequences using the same methods as with the birdsong and speech data. We also fit a power-law, exponential, and composite model to the MI decay, in the same manner (Fig. 2).

A Markov model is a sequential model in which the probability of transitioning to a state ($x_n$) is dependant solely on the previous state ($x_{n-1}$). Sequences are generated from a Markov model by sampling an initial state, $x_0$ from the set of possible states $S$. $x_0$ is then followed by a new state from the probability distribution $P(x_n|x_{n-1})$. Markov models can thus be captured by a Matrix $M$ of conditional probabilities $M_{ab} = P(x_n = a|x_{n-1} = b)$, where $a \in S$ and $b \in S$. In the example (Fig. 2b) we produce a set of 65,536 ($2^{16}$) sequences from Markov models describing two Bengalese finches[31,32].

The hierarchical model from Lin and Tegmark[3] samples sequences recursively in a similar manner to how the Markov model samples sequences sequentially. Specifically, a state $x_0$ is drawn probabilistically from the set of possible states $S$ as in the Markov model. The initial state $x_0$ is then replaced (rather than followed by, as in the Markov model) by $q$ new states (rather than a single state as in the

Markov model), which are similarly sampled probabilistically as $P(x_i|x_0)$, where $x_i$ is any of the new $q$ states replacing $x_0$. The hierarchical grammar can therefore similarly be captured by a conditional probability matrix $M_{ab} = P(x_{l+1} = a|x_l = b)$. The difference between the two models is that the sampled states are replaced recursively in the hierarchical model, whereas in the Markov model they are appended sequentially to the initial state. In the example (Fig. 2a) we produce a set of 1000 sequences from a model parameterized with an alphabet of 5 states recursively subsampled 12 times, with 2 states replacing the initial state at each subsampling (generating sequences of length 4096).

The final model combines both the Markov model and the hierarchical model by using Markov-generated sequences as the end states of the hierarchical model. Specifically, the combined model is generated in a three-step process: (1) A Markov model is used to generate sequences equal to the number of possible states of the hierarchical model (S). (2) The combined model is sampled in the exact same manner as the hierarchical model to produce sequences. (3) The end states of the hierarchical model are replaced with their corresponding Markov-generated states from (1). In the example (Fig. 2c) we produce sequences in the same manner as the hierarchical model. Each state of these sequences is then replaced with sequences between 2 and 5 states long generated by a Markov model with an alphabet of 25 states.

Neither the hierarchical model nor the combined model is meant to exhaustively sample the potential ways in which hierarchical signals can be formed or combined with Markovian processes. Instead, both models are meant to illustrate the theory proposed by prior work and to act as a baseline for comparison for our analyses on real-world signals.

**Reporting summary**. Further information on research design is available in the Nature Research Reporting Summary linked to this article.

## Data availability

The European starling song data set is available on Zenodo[64]. The availability of the California thrasher, Cassin's vireo, and Bengalese finch data sets are at the discretion of the corresponding laboratories and are currently publicly hosted at Bird-DB[40] and FigShare[33,63]. The Buckeye (English)[36], GECO (German)[37], and AsiCA[38] (Italian) speech corpora are currently available for research purposes through their respective authors. The CSJ[39] (Japanese) corpus is currently available from the authors for a fee upon successful application. A reporting summary for this article is available as a Supplementary Information file.

## Code availability

The software created for all of the analyses performed in this article are available at https://github.com/timsainb/ParallelsBirdsongLanguagePaper. The tools used for building the European starling corpus are available at https://github.com/timsainb/AVGN.

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

## Acknowledgements

We thank David Nicholson, Richard Hedley, Martin Cody, Zeke Arneodo, and James Jeanne for making available their birdsong recordings to us. Work supported by NSF Graduate Research Fellowship 2017216247 to T.S., and NIH R56DC016408 to T.Q.G.

## Author contributions

T.S. and T.Q.G. devised the project and the main conceptual ideas. T.S. carried out all experiments and data analyses. T.S. and T.Q.G. wrote the paper. T.S., B.T., M.T., and T.Q.G. were involved in planning the experiments, and contributed to the final version of the paper.

## Additional information

**Competing interests:** The authors declare no competing interests.

