## [Peer Review File · Nature Communications]

Reviewers' Comments:

Reviewer #1:

Remarks to the Author:

In this manuscript, "Parallels in the sequential organization of birdsong and human speech, Sainburg et al. ask a question that is both conceptually and methodologically interesting: if a bird or human makes a sound, how much information do we have about what sounds might come next, and after that, and after that? To address this, the authors explore the information decay across the songs of three species of birds and the language of English speakers in Ohio.

I think this paper could be revised to be informative for the broad readership of Nature Communications, but in its current form I found it to be difficult to parse. I am probably in the 'target audience' of this paper since I study both birdsongs and human languages, and I still found myself puzzling over the hypotheses, methods, and results.

First, terms and assumptions need to be defined in a clear way at their first occurrence in the text to specify the authors' meaning.

Mutual information

Regular grammar

Markovian process vs. non-Markovian process

Hierarchical structure vs. syntax

What is the null hypothesis here? I think more could be done to define and test the competing hypotheses. The abstract sets up two competing hypotheses: human syntax imposes hierarchical structure on language (i.e. syntax evolved first), vs. syntax developed to exploit existing hierarchical structure (i.e. hierarchical structure evolved first). First, I would like to see the distinction between the two made clear for the general scientific audience that does not study language. Second, I think it would be beneficial to explicitly show what you expect your plots to look like if syntax evolved first vs. hierarchical structure evolved first (or, if syntax existed without hierarchical structure vs. if hierarchical structure existed without syntax). I suggest that the authors make a synthetic dataset for both of these scenarios. In addition to clarifying the differences between the scenarios for the reader, seeing what the mutual information decay plots would look like in these two scenarios should make it so that the plots of the actual data visually demonstrate to the reader that one hypothesis is supported over the other. Under which circumstances would we expect a solely exponential, solely power-law, or concatenative plot?

As with any focal study, the three species of bird chosen here are not representative of all oscines, nor are American English speakers necessarily representative of all humans, so care is needed in generalizing to all song or speech. The authors present mutual information decay curves and suggest that the shapes of these curves are indicative of the underlying structure of the vocalizations and also associated with the syllable repertoire of the bird species or the phoneme inventory size of the language. I think this claim would be more convincing if it held for bird species that structure their songs very differently as well. Many species of bird have one song that they repeat over and over, or a finite number of song types, with a few syllables each. In these, I might expect the Markovian processes to dominate the signal in the mutual information decay plots, which would be a good proof-of-concept for the authors' experiment. At the far extreme are birds like chipping sparrows, in which one could predict with relative certainty which syllable comes after any given syllable. I know that zebra finch corpora are available (e.g. http://ofer.sci.ccnycunyu.edu/song_database); for other species it might be easier to find a long recording on Macaulay library, xeno-canto, etc.

Similarly, the authors' imply that their results for human language are linked to the number of

phonemes in that language, but that result would be much more salient if there were one or more additional languages for comparison. Would the results change not only based on the number of phonemes but also based on their organization? For example if two languages had the same number of phonemes but one had a consonant-vowel-consonant-vowel organization like Japanese or Hawaiian, would the mutual information decay be different?

The study of genomes and music at the end appears tacked on, without much explanation of the different shapes of the curves or the expectation for each data type.

Specific points:

I think the authors conduct most analyses at the syllable level in songs and the phoneme level in languages. The authors could add clarification about the levels of organization in sounds and language and justify this choice. I can envision an argument that syllables in songs are more analogous to words than to phonemes. For example, in Figure 1, if we look at syllables D, E, and F, it seems that syllable E is basically the first part of syllable D combined with the second part of syllable F. Phonemes, in contrast, could not really be subdivided in this way. I am basically fine with the authors' choice here, but the levels of organization could be better explained.

Line 13 – “Most theories assume” -- this phrasing led me to expect more than one reference.

Lines 211-217 – how often are syllables resized vs. padded?

Lines 246-250 – here the authors train the model on individual starlings and pool across individuals for Bengalese finches and Cassin's vireo. What are the effects of this methodological choice? What happens if you do the opposite, pooling starlings and separating the other two by individual?

Equations 4 and 5 – without knowing exactly what the authors put into the digamma function, the equation for $S_{\hat{}}$ seems to be summed over the whole repertoire. So in equation 5, how exactly are the marginal and joint entropies for a pair of syllables calculated?

I found figure 2 difficult to parse – some of the panels are too small for me to draw my own conclusions about the data. As I mentioned, I'm not entirely sold on the idea of analyzing individual starlings and pooling other species.

Figure 4 – how are the authors subtracting out the power-law component vs. the exponential component, and what assumptions need to be made in that process?

Extended data Figure 1, 9 – it would be helpful to have the species name on top of each plot in addition to the individual bird identifier.

Extended data Figure 3. Is the x-axis measured in syllables in all plots? The caption seems to measure the minima in phonemes, words, or parts of speech.

Reviewer #2:

Remarks to the Author:

Review of Sainburg et al

This paper presents an analysis comparing the scaling of mutual information (MI) with distance in

human speech and birdsong of three different passerine species. For most of the data from both humans and birds, the best model to explain the relationship between element distance and MI combines an exponential decay at short distances with a power-law decay at longer distances. The authors conclude that their data support a “common model for vocal syntax, shared among multiple species” and speculate that this is informative about the evolution of human language.

From a methodological viewpoint, this paper is a tour-de-force, combining large datasets with state-of-the-art methods in “deep learning” (convolutional autoencoding) to allow mostly automated analysis that would be unthinkable based on traditional, hand-annotated birdsong corpora. However, I think the authors currently undersell this approach by making speculative connections to language evolution that seem to me both unconvincing and unnecessary. The methods and findings are important and stand on their own without such speculation, which in my opinion distracts from the central value of this new approach (as I detail below).

Thus, I think this paper should definitely be published, but only after being rewritten so as to focus on its methodological and empirical strengths and to decrease or eliminate the speculation about language evolution.

Detailed comments

I first explain why I think the “language evolution” sales pitch adopted in the current ms. is inappropriate. First, the statement in the abstract that “Evidence of hierarchical structure in non-human vocalizations would support the ... hypothesis” that “syntax may have developed to exploit a hierarchical structure already present in proto-linguistic communication signals” because (from the conclusion) “human proto-languages shared similar hierarchical dependencies” is not necessarily true. It would be correct if hierarchical structure was found in a species or clade closely related to humans, so that we could conclude such structure was present in a recent common ancestor’s proto-language. But in distant relatives like birds, who evolved the vocal learning capabilities underlying song independently of those underlying human speech, this inference doesn’t follow. Birdsong hierarchicality, if not shared with systems of primate communication, may tell us nothing at all about the phylogenetic sequence of human language evolution, but rather be consistent with some more global constraints on the generation of complex sequences, followed by both humans and birds (and perhaps humpback whales as well, see Suzuki et al 2006, which should be more prominently cited, i.e. in the main text).

Second, the authors examine MI between phonemes in humans and syllables in birds, neither of which is necessarily relevant to phrasal syntax (which involves words and other multi-phoneme and often multi-syllabic morphemes). So they are really comparing apples, oranges, and bananas here. I would be more comfortable with (but would not demand) a syllable-based analysis in the speech data, and a complete omission of comparisons with word-based hierarchy. This whole comparison feels like a weak foundation upon which to present this impressive empirical research.

A second issue concerns the definition and measurement of “hierarchy”. The authors blithely state that power law dependencies are “thought to reflect the hierarchical syntactic structures between words”, citing three papers by Wentian Li, two of which are unpublished and the remaining one published in a physics journal. It may be correct to say “thought by Wentian Li to reflect...”, but the current wording suggests that this is widely agreed (e.g. among computational linguists or speech engineers) which to my knowledge is not the case. Again, this is unconvincing, and again unnecessary: the similarities between the models for speech and birdsong are interesting in their own right, without reference to language syntax or meaning. There is plenty of work out there (e.g. that of Ferrer i Cancho) which is both more relevant to the current paper and less speculative.

Further regarding “hierarchy”, an unexamined alternative possibility what the authors see in the human speech data may be a reflection of duality of patterning (the fact that language has two levels, one combining meaningless phonemes into words, and another combining meaningful words into sentences) – in itself interesting, but not obviously relevant to birdsong. Although I am happy to term duality of patterning “hierarchical”, this is not the same sense as used by linguists to refer to syntactic hierarchy.

Finally, I think the idea that phonology (roughly, word structure) requires only sub-regular grammars while phrasal syntax (roughly, sentence structure) requires supra-regular grammars is an interesting one, and worth pursuing more here. At least, the recent work from Hienz & Idsardi should be cited in this regard; there is also some brand new work concerning this about monkeys that might be mentioned (Wang lab, see also a commentary by Fitch). But the problem with all of this, as already suggested above, is that the authors’ human analysis is NOT of phrasal syntax! Indeed, to the extent that they’re really analysing phoneme-level dependency we don’t even expect supra-regularity to apply if these authors are correct. I of course recognize that any word-level dependencies would have to be reflected in phoneme-level dependencies (since words are made of phonemes), but this is not really the appropriate analysis to reveal such higher-level dependencies.

I also think the authors might devote some space to the differences between the distributions illustrated in Fig 4, which to my eye seem pretty similar among the birds and pretty different for humans (again possibly because these compare phoneme distances and word lengths with syllable differences and bout lengths). What do the differences in the parameterization of the exponential and power law components mean?

As a small point, there are plenty of misspelled words and the reference list is both incomplete and incorrectly capitalized throughout (e.g. dna, “science” for the journal, or *cebus olivaceus*). Please fix this in the resubmission.

1. Ferrer i Cancho, R. and R.V. Solé, The small world of human language. *Proceedings of the Royal Society B*, 2001. 268: p. 2261-2265.
2. Ferrer i Cancho, R. and R.V. Solé, Least effort and the origins of scaling in human language. *Proceedings of the National Academy of Sciences, USA*, 2003. 100: p. 788-791.
3. Gustison, M.L., et al., Gelada vocal sequences follow Menzerath’s linguistic law. *Proceedings of the National Academy of Sciences*, 2016. 113(19): p. E2750–E2758.
4. Heinz, J. and W. Idsardi, Sentence and word complexity. *Science*, 2011. 333: p. 295-297.
5. Heinz, J. and W. Idsardi, What complexity differences reveal about domains in language *Topics in Cognitive Science*, 2013. 5: p. 111-131.
6. Fitch, W.T., Bio-Linguistics: Monkeys Break Through the Syntax Barrier. *Current Biology*, 2018. 28(12): p. R695–697.
7. Jiang, X., et al., Production of supra-regular spatial sequences by macaque monkeys. *Current Biology*, 2018.
8. Pulleyblank, E.G., The meaning of duality of patterning and its importance in language evolution, in *Studies in Language Origins*, J. Wind, et al., Editors. 1989, Benjamins: Amsterdam. p. 53-65.

Reviewer #3:

Remarks to the Author:

This manuscript presents what would appear to be a very interesting finding: that the mutual information in different communicative sequences appears to be under the control of different processes at short range and at long range within the sequence. This result could be of interest to a wide range of readers, but the manuscript suffers from a couple of very serious flaws that would need to be addressed before it would be possible to give a comprehensive assessment of its importance.

Firstly (and less critically), the manuscript is a confusing mix of two completely separate research projects. On the one hand, the "interesting" findings outlined in the abstract, and on the other, a novel automatic method for segmenting and annotating birdsong syllables. Here is the problem: the authors have developed a clever and sophisticated algorithm that presumably performs better than existing methods, and need to present that algorithm to the reader, as all the data analysed derives from their novel technique. But the paper itself appears to be about birdsong syntax, not machine learning. Nonetheless, the reader is given large amounts of information about the implementation of the encoder, and little information on the background to the syntactic hypotheses. I'm not sure there's a simple solution to this, as it is, in fact, important to present the novel methodology; but it does detract substantially from the main message of the manuscript. One possibility is to publish the methodology separately and reference it from this manuscript, but that clearly would delay publication substantially. Another possibility is to mention the new methodology with only brief detail, but to compare the results to a standard existing method for coding syllables. This would give the reader some confidence that, although we do not understand the method the authors used, it is broadly comparable to (but better than!) methods that we do understand.

Whatever solution the authors choose, the current presentation is not really acceptable. The somewhat handwaving introduction to the methodology from line 56 onwards does not give the reader confidence. Even with the (excessive) detail of the autoencoder given in the Methods, statements like, "in a manner akin to multidimensional scaling" (line 64) are neither clear enough to the non-expert, nor specific enough for the expert. Similarly, the use of specialist terminology ("stride", "nonlinearity") without explanation helps no one. Presenting equations like Equation 2 is also unhelpful unless the context is explained more – and that would push the manuscript to be even more methodological, rather than focussing on the findings.

The second problem is that the manuscript doesn't really present a clear hypothesis, nor explain how the authors' findings deviate from what might be expected under some (non-presented) null hypothesis. For example, the Mutual information decay fitting (line 310) is a very nice idea, and one that I think probably reveals important results, but the manuscript is lacking any theoretical background to place this in context. Why are you proposing a concatenation of these two particular models? More importantly, how would you expect the MI to decay under different generational models (a null hypothesis)? The authors need to present at least a summary of the MI decay expected under Markovian and various non-Markovian processes, otherwise how can we judge whether the results are unexpected or not? This question is fleetingly addressed on lines 87-90, and genuinely caught my interest! But it's simply not developed enough, and this really should be expanded, clearer, and the main thrust of the manuscript, rather than the detail of the neural network. On line 114 the authors state that exponential MI decay indicates a Markov process, but as this is absolutely central to the manuscript, this needs to be explained in much more detail. Indeed, the main result (lines 97-98) and the main hypothesis (lines 120-124) are buried in the text, rather than highlighted.

Another thing that rankled me right from the outset (although it really shouldn't have done so), was the way that the authors used the terms "distance" and "decay". The abstract (line 12), and the very start of the introduction (line 31) gives no explanation for what this means, and a reader could easily think that the manuscript is about geographical distance between birds! The text moves immediately

on to talk about "strength" (line 35), but doesn't say what this means – mutual information is not mentioned until the following sentence. The central idea of "decay in MI" isn't explained until line 72. I think that the manuscript needs to be rewritten, not from the point of view of an information theory expert, but with a more general reader in mind.

I also have one more technical question. On line 106 the authors say that the transition points were close to the bout length. If these bouts are separated by long silences, then they may be effectively different "messages", and so have different statistical properties. Are we therefore surprised by this finding? This should be mentioned and explored.

I also have a number of more general points:

Line 31: Language must make use of, or empirically does make use of?

Lines 38-40: Introduce the hierarchy of generative grammars for those readers who are unfamiliar.

Line 67: I would like to see the accuracy of the validation presented here.

Line 76: The speech dataset hasn't been mentioned before. Introduce it before referring to it.

Line 139: I don't know what this means.

Lines 144-161: This is such a diverse set of sources that you should discuss the implication in the text. Some are wild, some captive (and captive reared?). Although I don't think it necessarily detracts from the conclusions, it's still important to give some discussion.

Line 191: What is your justification for this? The whole segmentation process is quite novel and a little quirky, so (if this manuscript were focussed on the methodology) it would be good to give some graphs to illustrate.

Response to referees

Major revisions, reflecting the concerns of multiple reviewers, are described in the first section of this response. Following the major revisions, we provide a point-by-point response to each of the reviewers' questions and concerns.

Major revisions

01. Reviewers noted problems with the framing of our paper in the context of language evolution. We agree with this critique and have rewritten the manuscript to focus on the central hypotheses that hierarchical and Markovian processes are reflected in the organization of both speech and birdsong.
02. As suggested by Reviewer #3, the manuscript now emphasizes the results of our information theoretic analyses rather than the algorithmic approaches used to generate datasets. In place of the elaborate neural-network based approach in the original ms., we now present results based on hand-transcribed datasets for Cassin's vireo, Bengalese finch, and a new species, the California Thrasher. We use our segmentation algorithm (as described in the original ms) for the European starling song data. To simplify our methods, we now employ an established unsupervised dimensionality reduction and clustering pipeline, in the place of our more complicated neural network. This new approach yields empirically identical results as reported in the original manuscript, and is much simpler to describe. Most of the discussion of the segmentation algorithm has been moved to the methods.
03. All three reviewers noted the need for better articulation of the explicit hypotheses we are testing. To address this, we added a "computational models" section to the methods in which we outline our null hypotheses for the expected decay in Mutual Information for signals generated by Markovian and hierarchical processes. We also revised our justification for the choice of these two models, in the context of current theories about speech and birdsong, and added a figure (Figure 2) that gives a schematic representation of each model along with the shape of its associated MI decay function.
04. As suggested by Reviewer #1, we have expanded the number of datasets from the original study. For speech, we added German, Japanese, and Italian datasets to our analysis, replicating our original findings (for English) in each. For birdsong, we considered adding Canary and Zebra Finch songs to the revised manuscript (see response 1.4 below), but ultimately added only one species of songbird, the California Thrasher which is hand-transcribed in a manner similar to the Cassin's vireo.

Note: In the point-by-point rebuttal that follows, we have highlighted the full text of the reviewers' original comments. Our response to each specific question or concern is shown in italics.

Reviewer #1 (in green):

In this manuscript, “Parallels in the sequential organization of birdsong and human speech, Sainburg et al. ask a question that is both conceptually and methodologically interesting: if a bird or human makes a sound, how much information do we have about what sounds might come next, and after that, and after that? To address this, the authors explore the information decay across the songs of three species of birds and the language of English speakers in Ohio.

I think this paper could be revised to be informative for the broad readership of Nature Communications, but in its current form I found it to be difficult to parse. I am probably in the ‘target audience’ of this paper since I study both birdsongs and human languages, and I still found myself puzzling over the hypotheses, methods, and results.

- 1.1. First, terms and assumptions need to be defined in a clear way at their first occurrence in the text to specify the authors' meaning.
 - 1.1.1. Mutual information
We added an explanation aimed at the general public when mutual information is first mentioned (line 35-37). Additionally, mutual information is defined in the Methods section (line 321-336).
 - 1.1.2. Regular grammar
We rewrote the first mention of regular grammars to indicate that regular grammars are underlied by Markov processes (line 127).
 - 1.1.3. Markovian process vs. non-Markovian process
See 1.1.2
 - 1.1.4. Hierarchical structure vs. syntax
We have re-written the introduction to more clearly describe the general hierarchical organization in natural languages, of which phrasal syntax is only one part (lines 30-44). In addition, in the final paragraph of the main text (line 202-214) we differentiate between hierarchical organization and syntactic organization which might underlie language.
- 1.2. What is the null hypothesis here? I think more could be done to define and test the competing hypotheses. The abstract sets up two competing hypotheses: *human syntax imposes hierarchical structure on language (i.e. syntax evolved first)*, vs. *syntax developed to exploit existing hierarchical structure (i.e. hierarchical structure evolved first)*. First, I would like to see the distinction between the two made clear for the general scientific audience that does not study language
We now make an explicit statement of our hypotheses in paragraph four of the introduction (line 103):

Because separate aspects of natural language can be explained by Markovian and non-Markovian dynamics, we hypothesized the MI decay observed in human language should be best explained by a pattern of MI decay similar to that observed in the artificial concatenative model, which combines both Markovian and hierarchical processes. Likewise, we hypothesized that Markovian dynamics alone would not provide a full explanation of the MI decay in birdsong.

In addition, we added a section, titled “computational models” to the methods, where we explain the expected MI decay for each of our competing hypotheses (Markovian, hierarchical, concatenative). These models and their predicted MI decay are also now reviewed in Figure 2.

- 1.3. *Second, I think it would be beneficial to explicitly show what you expect your plots to look like if syntax evolved first vs. hierarchical structure evolved first (or, if syntax existed without hierarchical structure vs. if hierarchical structure existed without syntax). I suggest that the authors make a synthetic dataset for both of these scenarios. In addition to clarifying the differences between the scenarios for the reader, seeing what the mutual information decay plots would look like in these two scenarios should make it so that the plots of the actual data visually demonstrate to the reader that one hypothesis is supported over the other. Under which circumstances would we expect a solely exponential, solely power-law, or concatenative plot? In light of requests from reviewers 2 and 3 we have removed our discussion of the evolutionary origins of our observations. Beyond this, the reviewer’s suggestion to describe more clearly the MI decay predicted by the Markovian and non-Markovian processes is warranted. As noted above (1.2), we now do this explicitly in the new section on “computational models” (line 394) and in Figure 2.*
- 1.4. *As with any focal study, the three species of bird chosen here are not representative of all oscines, nor are American English speakers necessarily representative of all humans, so care is needed in generalizing to all song or speech. The authors present mutual information decay curves and suggest that the shapes of these curves are indicative of the underlying structure of the vocalizations and also associated with the syllable repertoire of the bird species or the phoneme inventory size of the language. I think this claim would be more convincing if it held for bird species that structure their songs very differently as well. Many species of bird have one song that they repeat over and over, or a finite number of song types, with a few syllables each. In these, I might expect the Markovian processes to dominate the signal in the mutual information decay plots, which would be a good proof-of-concept for the authors’ experiment. At the far extreme are birds like chipping sparrows, in which one could predict with relative certainty which syllable comes after any given syllable. I know that zebra finch corpora are available (e.g. http://ofer.sci.cuny.cuny.edu/song_database); for other species it might be easier to find a long recording on Macaulay library, xeno-canto, etc.*

We agree with the reviewer that we should be careful about generalizing from the small number of species analyzed in the original paper. We now make this point in the revised manuscript (lines 196-199). Given the novelty of the observed similarity between human language and vocalizations in any non-human species, it is natural to wonder how general it may be. To help address this question, we have followed the reviewer’s suggestion and added additional songbird species and human language analyses to the revised manuscript.

We searched for additional birdsong corpora that met the criteria needed for our current analyses: (1) large datasets containing hundreds/thousands of bouts/songs from multiple individuals, and (2) Either hand transcribed labels or clean recordings from sound isolated environments that would allow us to perform automated computational segmentation and transcription. We found three potential candidate song datasets: (1) a dataset of syllabically transcribed wild recordings of California Thrasher song (Sasahara et al. 2012; Cody et al. 2016), (2) a large-scale dataset of isolated Zebra finch recordings (Pearre et al. 2017), and (3) a dataset of hand-transcribed Canary song (Markowitz et al. 2013), which is transcribed phrasally rather than syllabically. Of these, we ultimately decided to include only the California Thrasher song in our analyses, for the reasons listed below.

Zebra Finch: The zebra finch dataset was large scale in that thousands to tens of thousands of syllables were produced by each individual bird. The primary issue with the dataset is that the songs were pre-segmented into (as the reviewer mentions) the single motif, a standard unit for Zebra finch vocalization studies. This is not ideal for characterizing the structure of Zebra finch vocalizations, however, as the segmentation discards any potential inter-motif information. Recent characterizations of Zebra finch song show complex sequential organization including non-adjacency relationships in inter-motif syllables (Hyland Bruno and Tchernichovski 2017). When we run our analysis on this dataset, we observe that the MI decay of Zebra finch song is dominated by an exponential decay, but in most cases (5 of 6 birds) was best-fit by a concatenative model. In contrast to what we observed in other species (Fig 4c), however, residuals from the exponential fit do not appear to follow a power-law (Fig. R1b,d). One might infer from the absence of a strong power-law component that, as the reviewer suggested, zebra finch song does not contain any hierarchical dynamics. Alternatively, the failure to see a power-law may be due to the missing inter-motif information. Because there is reasonable doubt about the source of the structure of the signal we think it is best not to include the Zebra finch dataset in the current analysis.

Fig R1. (A,C) MI decay of Zebra finch song. (B,D) Residual decay after subtracting the exponential component. Note the absence of a clear power law, in contrast to Figure 4 in the revised ms. Model fit is in orange, MI decay data is in blue. Fit model components in (A,C) are dotted (power-law) and dashed lines (Markov model).

Canary: The canary dataset we considered using was the same as that used in a published paper to show high-order Markovian relationships in canary song (Markowitz et al. 2013). The dataset is hand transcribed at the level of phrases, which comprise successive repetitions of the same (or very similar) syllables. Each transcribed bout is comprised of ~5-20 phrases. Despite the disparity in the level of analysis (phrase vs. syllable) we looked at the mutual information decay of the data and observed that across birds, the MI decay was consistently well fit by an exponential decay alone (4 of 6 birds). Again, this may reflect the absence of hierarchical dynamics in canary song governing the sequencing of phrases or may reflect the loss of information contained within phrases, or the short (5-20 element) sequence lengths. We tried to employ the algorithmic syllable segmentation as described in the manuscript for starling song, but couldn't produce a satisfying parsing of the phrases, and it became clear that because no clear silences exist between syllables, we would need to substantially modify our existing segmentation methods. After several failed attempts it was clear that accurate automated transcription of this canary song dataset would be a full bioacoustics project in itself, and was beyond the scope of the present manuscript. We are continuing to work on a novel solution to this problem, but this method will not be ready in time for the completion of this manuscript. Because we could not look at the sequential organization of the canary song bouts in a comparable method to our other species datasets, interpreting the absence of a power-law component, as with the zebra finch data, is difficult. Because of this ambiguity, we elected not to include the canary song analysis in the revised manuscript.

Figure R2: (A) Waveform (top), spectrogram (middle), and phrase labels (blue vertical lines) of a single bout of canary song. In red bounding boxes are two different renditions of what is labeled as the same phrase show a large variation of syllables within phrase. (D) Observed mutual information decay of two individuals.

California Thrasher: The thrasher song is recorded in the wild from a group of focal individuals. There are up to several thousand syllables from each individual, but fewer songs

(total and per individual) than are available for the Cassin's vireo. We include it as an additional dataset. The structure of the thrasher song follows the same general form as the three species in the original ms. These new data appear alongside the other species in Figure 4. We note that we had considered including this dataset into our original manuscript, but decided against it because of its somewhat smaller size and because the song syllables did not cluster as cleanly as those for the other species (see Figure 1 in the revised manuscript). Because our analysis is now on hand-transcriptions, the automated clustering is no longer a problem and so we can include it as an additional dataset.

We are sensitive to the concern that we are only selecting those species for inclusion who's songs follow the same general form as that observed for speech. This is not our goal. We recognize that there may be many species of songbirds (and other animals) that do not produce sequential vocal communications signals that are governed by both Markovian and hierarchical processes. As such, we are happy to include either (or both) the canary and zebra finch datasets even though their results are ambiguous. For the purposes of the present study, however, we think the most noteworthy result is the fact that any non-human species show this organization, and that our current set of results makes this point convincingly. Exploring the generality of these dynamics across a wider range of songbird species (and other taxa) where the datasets allow one to rule out the absence of a given decay will be important for future work.

- 1.5. Similarly, the authors' imply that their results for human language are linked to the number of phonemes in that language, but that result would be much more salient if there were one or more additional languages for comparison. Would the results change not only based on the number of phonemes but also based on their organization? For example if two languages had the same number of phonemes but one had a consonant-vowel-consonant-vowel organization like Japanese or Hawaiian, would the mutual information decay be different?

It was not our intent in the original manuscript to link the results for human language to the number of phonemes. We have revised the manuscript to be sure that we avoid this implication. We do however agree with the reviewer that analysis of additional languages would strengthen our claims. As the reviewer notes, phonological organization differs between languages and our results in English may differ from those seen in other languages. We were able to find three more linguistic datasets of similarly transcribed spontaneous speech of a similar duration to that in the Buckeye corpus. The new datasets are in German, Italian, and Japanese (although the phonetic alphabets and methods of transcription differ between datasets). Our original results with English are replicated in these new datasets (see Fig. 3). In addition, we also analyzed the MI decay in the language datasets when they were parsed according to word, syllable, mora, and part-of-speech transcriptions where available for each dataset. In all cases, the MI decay is best fit by a concatenative model, except in the case of German words, where the MI decay is best fit by a power-law (Extended Data Fig. 1).

- 1.6. The study of genomes and music at the end appears tacked on, without much explanation of the different shapes of the curves or the expectation for each data type.

We agree. The MI decay of music and genomes have been previously published (Lin and Tegmark 2016; Li and Kaneko 1992; Levitin et al. 2012), and our analysis was only meant for reference. We have removed this section.

- 1.7. I think the authors conduct most analyses at the syllable level in songs and the phoneme level in languages. The authors could add clarification about the levels of organization in sounds and language and justify this choice. I can envision an argument that syllables in songs are more analogous to words than to phonemes. For example, in Figure 1, if we look at syllables D, E, and F, it seems that syllable E is basically the first part of syllable D combined with the second part of syllable F. Phonemes, in contrast, could not really be subdivided in this way. I am basically fine with the authors' choice here, but the levels of organization could be better explained.

We performed our primary analyses on phonemes in language datasets (Fig 3) and syllables in song datasets (Fig 4) because these units were most common across datasets. We have added wording in the revised manuscript (lines #216-224) to better clarify and justify our choice for the primary analysis unit. Where available in the language datasets, we performed additional analyses on syllables (German), part-of-speech (English, Japanese), mora (Japanese), and words (German, Japanese, English). These results (Extended Data Fig. 1) are consistent with those in phonemically-parsed speech (Fig. 3), and are consistent with prior work indicating that long-range correlations flow across levels in linguistic datasets, at least from words to characters (Altmann et al. 2012). The choice of the "right" unit for birdsong (or other non-human communication signals) is somewhat arbitrary, and the term "syllable" can refer to units at different levels in the acoustic hierarchy between species. We tried to avoid this problem by using the unsupervised parsing and clustering methods described in the original paper. In the end, we chose to look primarily at syllables for the birdsong datasets because these are typically the only form of transcriptions available, and tend to be the simplest to transcribe. In Extended Data Fig. 9 (Extended Data Fig. 8 in the original ms), we show the results of our MI analysis on segmentations in the raw-audio at three different temporal scales (0.01s, 0.1s, 1s). All of these show MI decay dynamics that are similar to that shown (Fig 4) for syllables.

- 1.8. Line 13 – "Most theories assume" -- this phrasing led me to expect more than one reference. ' This sentence was removed from the current revision of the manuscript.
- 1.9. Lines 211-217 – how often are syllables resized vs. padded?
This only affects the starlings (in the current revision). Approximately 5.8% of the motifs are resized versus padded. This is equal to the percentage of syllables that are longer than the threshold of 1 second.
- 1.10. Lines 246-250 – here the authors train the model on individual starlings and pool across individuals for Bengalese finches and Cassin's vireo. What are the effects of this methodological choice? What happens if you do the opposite, pooling starlings and separating the other two by individual?
This was an artifact of the way that the networks in the original paper were designed and trained. Since we are no longer using the networks for dimensionality reduction, the reviewer's

point is not relevant to the revision. To answer the question, however, we can empirically test any effects of pooling vs separating individuals. In the handful of birds that we tried this with when developing the method, we found no difference in clustering quality in Bengalese finches, and an improvement in clustering quality in Cassin's vireo. These effects are likely due to the high inter-individual overlap in syllable repertoire across the Cassin's vireo dataset, which leads to more exemplars when the data are pooled across singers. Conversely, individual Starlings have very little overlap in the syllables they produce, and there is little gain in clustering quality when pooling across individuals. We intend to address this question more rigorously in a follow-up paper, now in preparation, that describes the network-based dimensionality reduction approach that appeared in the original ms.

- 1.11. Equations 4 and 5 – without knowing exactly what the authors put into the digamma function, the equation for $S_{\hat{}}$ seems to be summed over the whole repertoire. So in equation 5, how exactly are the marginal and joint entropies for a pair of syllables calculated?

We clarified the mutual information estimation section to give a better example of MI calculation on a sequence, better explaining the distributions X and Y and their joint distribution, as well as how entropy is calculated over those distributions (lines #321-336). In addition, full code will be made available upon publication. Below is the function for calculating MI using the methods described in the original and revised manuscript. The results are empirically equivalent to a naive calculation of entropy. The function is based upon the SciPy function for efficiently calculating MI naively.

```

from scipy import sparse as sp
import numpy as np
import scipy.special
def est_mutual_info(a,b):
    # contingency matrix of a * b
    contingency = contingency_matrix(a, b, sparse=True)
    nzx, nzy, Nall = sp.find(contingency)
    # entropy of a
    Na = np.ravel(contingency.sum(axis=0)) # number of A
    S_a = entropy(Na) # entropy with P(A) as input
    # entropy of b
    Nb = np.ravel(contingency.sum(axis=1))
    S_b = entropy(Nb)
    # joint entropy
    S_ab = entropy(Nall)
    # mutual information
    MI = S_a + S_b - S_ab
    return MI

def entropy(Nall):
    N = np.sum(Nall)
    pAll = np.array([float(Ni) *
scipy.special.psi(float(Ni) for Ni in Nall))
    S_hat = np.log2(N) - 1. / N * np.sum(pAll)

```

```
return S_hat
```

- 1.12. I found figure 2 difficult to parse – some of the panels are too small for me to draw my own conclusions about the data. As I mentioned, I'm not entirely sold on the idea of analyzing individual starlings and pooling other species.
We removed Figure 2. Both the Cassin's and the Bengalese finch datasets are now hand-transcribed which should resolve this issue.
- 1.13. Figure 4 – how are the authors subtracting out the power-law component vs. the exponential component, and what assumptions need to be made in that process?
The subtracted components are simply the "signal" minus the "modeled component" (e.g. power-law or exponential) and the modeled y-intercept. This assumes that the fit model is a linear combination of the two components, such that subtracting one component (e.g. power-law) should leave the other component (e.g. exponential) plus the residual noise.
- 1.14. Extended data Figure 1, 9 – it would be helpful to have the species name on top of each plot in addition to the individual bird identifier.
We added a color-key to each figure caption.
- 1.15. Extended data Figure 3. Is the x-axis measured in syllables in all plots? The caption seems to measure the minima in phonemes, words, or parts of speech.
- 1.16. *This was a typo, thank you for pointing it out. The x-axis of each of our supplemental figures is now labelled with the correct units.*

2. Reviewer #2 (in blue):

This paper presents an analysis comparing the scaling of mutual information (MI) with distance in human speech and birdsong of three different passerine species. For most of the data from both humans and birds, the best model to explain the relationship between element distance and MI combines an exponential decay at short distances with a power-law decay at longer distances. The authors conclude that their data support a "common model for vocal syntax, shared among multiple species" and speculate that this is informative about the evolution of human language.

From a methodological viewpoint, this paper is a tour-de-force, combining large datasets with state-of-the-art methods in "deep learning" (convolutional autoencoding) to allow mostly automated analysis that would be unthinkable based on traditional, hand-annotated birdsong corpora. However, I think the authors currently undersell this approach by making speculative connections to language evolution that seem to me both unconvincing and unnecessary. The methods and findings are important and stand on their own without such speculation, which in my opinion distracts from the central value of this new approach (as I detail below).

- 2.1. Thus, I think this paper should definitely be published, but only after being rewritten so as to focus on its methodological and empirical strengths and to decrease or eliminate the speculation about language evolution.

We followed the reviewer's advice and no longer use language evolution to motivate our hypotheses or as a source for speculation.

- 2.2. I first explain why I think the “language evolution” sales pitch adopted in the current ms. is inappropriate. First, the statement in the abstract that “Evidence of hierarchical structure in non-human vocalizations would support the ... hypothesis” that “syntax may have developed to exploit a hierarchical structure already present in proto-linguistic communication signals” because (from the conclusion) “human proto-languages shared similar hierarchical dependencies” is not necessarily true. It would be correct if hierarchical structure was found in a species or clade closely related to humans, so that we could conclude such structure was present in a recent common ancestor's proto-language. But in distant relatives like birds, who evolved the vocal learning capabilities underlying song independently of those underlying human speech, this inference doesn't follow. Birdsong hierarchicality, if not shared with systems of primate communication, may tell us nothing at all about the phylogenetic sequence of human language evolution, but rather be consistent with some more global constraints on the generation of complex sequences, followed by both humans and birds (and perhaps humpback whales as well, see Suzuki et al 2006, which should be more prominently cited, i.e. in the main text).

As noted in our responses to the other reviews (1.3), we have removed statements about the evolutionary origins of hierarchical structure, and now frame our hypothesis in terms of “global constraints” on biological processes (introduction, paragraphs 2&3). We have added a more prominent citation to Suzuki et al. (2006) in the main text of the revised manuscript (line # 49), where we now expand on the implications of our results in terms of animal communication.

- 2.3. Second, the authors examine MI between phonemes in humans and syllables in birds, neither of which is necessarily relevant to phrasal syntax (which involves words and other multi-phoneme and often multi-syllabic morphemes). So they are really comparing apples, oranges, and bananas here. I would be more comfortable with (but would not demand) a syllable-based analysis in the speech data, and a complete omission of comparisons with word-based hierarchy. This whole comparison feels like a weak foundation upon which to present this impressive empirical research.

We have worked to clarify the important distinctions between potential sources of sequential organization that are tied to explicit hierarchical levels of language (see response 1.1.4), and we agree with the reviewer that it is tricky to know the best way to compare vocal units across species (see response 1.7). Our goal here is simply to look at the statistics of sequences of acoustically defined (or definable) elements in speech and birdsong. We agree that neither birdsong syllables (as we define them) or phonemes are necessarily relevant to phrasal syntax. We now note this important distinction in the revised manuscript (line #203), and have added citations (introduction, paragraph 1) to relevant literature discussing the origins of the observed long-range correlations present in written texts, and their relation to higher level structure such as semantic content (Altmann et al. 2012; Ebeling and Neiman 1995).

Importantly, the significance of our results does not rest on a link to phrasal syntax. At the same time, we do not think that the similarity in the general pattern of sequential

organization described in the paper is tied exclusively to any single level of acoustic unit. When we analyzed the MI decay for speech from multiple languages parsed by phonemes, part-of-speech, mora, and words, where available (see 1.7, and Extended Data Fig. 1), except for German words they all show the same patterns. This similarity extends as well to the syllable-based analysis of speech suggested by the reviewer (Extended data Fig. 1, center column). Likewise, when we analyzed MI decay for birdsong datasets parsed at different temporal scales, the pattern of results observed for the syllable-based analyses (Fig. 4) persists (Extended data Fig. 9). Regardless of how these units relate to phrasal syntax, the dynamics of the sequential patterning between them is consistent across many temporal scales and hierarchical levels in both speech and birdsong.

- 2.4. A second issue concerns the definition and measurement of “hierarchy”. The authors blithely state that power law dependencies are “thought to reflect the hierarchical syntactic structures between words”, citing three papers by Wentian Li, two of which are unpublished and the remaining one published in a physics journal. It may be correct to say “thought by Wentian Li to reflect...”, but the current wording suggests that this is widely agreed (e.g. among computational linguists or speech engineers) which to my knowledge is not the case. Again, this is unconvincing, and again unnecessary: the similarities between the models for speech and birdsong are interesting in their own right, without reference to language syntax or meaning. There is plenty of work out there (e.g. that of Ferrer i Cancho) which is both more relevant to the current paper and less speculative.

As noted above (see response 2.3), we now cite additional resources discussing the relationship between hierarchy and long-range correlations in natural language. Of these, Lin and Tegmark (2016) argue most strongly for the link between word-level syntax and the power-law decay of MI. At the same time, we appreciate the reviewer’s sense that our original wording was too strong in attributing power-law structure to word-level syntax, and have revised that section (lines #31-36). It now reads, “Similar relationships exist for the long-range dependencies between characters in texts, and are thought to reflect the hierarchical organization of natural language, where higher levels of abstraction (e.g., semantic meaning, syntax, words) govern organization in lower level components (e.g., parts-of-speech, words, characters)”. The relationship between a power-law and hierarchical organization is described in a range of literature outside of linguistics, as well as the work the reviewer mentions.

- 2.5. Further regarding “hierarchy”, an unexamined alternative possibility what the authors see in the human speech data may be a reflection of duality of patterning (the fact that language has two levels, one combining meaningless phonemes into words, and another combining meaningful words into sentences) – in itself interesting, but not obviously relevant to birdsong. Although I am happy to term duality of patterning “hierarchical”, this is not the same sense as used by linguists to refer to syntactic hierarchy.

As noted above, we agree with the reviewer that our discussion of hierarchy in the original manuscript was focused too restrictively on phrasal-syntax, and have revised the wording throughout the current manuscript to addresses this. Moreover, we have tried to avoid making specific predictions about the underlying source of hierarchical organization in the observed data. We agree that duality of patterning is potentially an alternative hypothesis for underlying

our results on the speech datasets but (likely) not the birdsong data, but prefer not to include speculative possibility in the paper.

- 2.6. Finally, I think the idea that phonology (roughly, word structure) requires only sub-regular grammars while phrasal syntax (roughly, sentence structure) requires supra-regular grammars is an interesting one, and worth pursuing more here. At least, the recent work from Hienz & Idsardi should be cited in this regard; there is also some brand new work concerning this about monkeys that might be mentioned (Wang lab, see also a commentary by Fitch).

We agree with the reviewer and would like to thank them for the references, many of which we now cite and discuss within the manuscript. We also more prominently emphasize our results and their relationship to prior literature such as Heinz and Idsardi in the main text (line #46).

- 2.7. But the problem with all of this, as already suggested above, is that the authors' human analysis is NOT of phrasal syntax! Indeed, to the extent that they're really analysing phoneme-level dependency we don't even expect supra-regularity to apply if these authors are correct. I of course recognize that any word-level dependencies would have to be reflected in phoneme-level dependencies (since words are made of phonemes), but this is not really the appropriate analysis to reveal such higher-level dependencies.

As the reviewer mentions, word-level dependencies should be reflected in phoneme level dependencies, as suggested by Altman et al., (2012). In addition, we perform our analysis on part-of-speech as well as words, where we find that in all analyses (except words in German) the exponential portion of the decay is still seen (although at shorter distances and to a lesser extent). The exponential portion of the decay might be expected in words, as phonological constraints can occur across words. The same observation in part-of-speech, however, suggests that other factors may also be at play and that the sequence dynamics we reveal here may not map directly onto exclusive levels of linguistic hierarchies. More work to understand these phenomena is required.

- 2.8. I also think the authors might devote some space to the differences between the distributions illustrated in Fig 4, which to my eye seem pretty similar among the birds and pretty different for humans (again possibly because these compare phoneme distances and word lengths with syllable differences and bout lengths). What do the differences in the parameterization of the exponential and power law components mean?

We agree with the reviewer and now emphasize these differences more strongly in the text (lines 142-149). This comment is perhaps more interesting in light of the new language data - where all languages show similar minima in curvature, that vary by only about 4 phonemes (~0.5 seconds). In contrast, for different species of songbirds the minima range between 2 syllables to tens of syllables (<1 second to tens of seconds). We note this variability in our results but cannot explain it. We speculate (in this rebuttal but not in the paper) that the high variability in birds reflects the broad phylogenetic distribution of the species studied, but this requires further study.

- 2.9. As a small point, there are plenty of misspelled words and the reference list is both incomplete and incorrectly capitalized throughout (e.g. dna, "science" for the journal, or cebus olivaceus). Please fix this in the resubmission.

We addressed all the grammatical errors we could find in the current revision.

3. Reviewer #3 (in red):

- 3.1. Firstly (and less critically), the manuscript is a confusing mix of two completely separate research projects. On the one hand, the “interesting” findings outlined in the abstract, and on the other, a novel automatic method for segmenting and annotating birdsong syllables. Here is the problem: the authors have developed a clever and sophisticated algorithm that presumably performs better than existing methods, and need to present that algorithm to the reader, as all the data analysed derives from their novel technique. But the paper itself appears to be about birdsong syntax, not machine learning. Nonetheless, the reader is given large amounts of information about the implementation of the encoder, and little information on the background to the syntactic hypotheses. I’m not sure there’s a simple solution to this, as it is, in fact, important to present the novel methodology; but it does detract substantially from the main message of the manuscript. One possibility is to publish the methodology separately and reference it from this manuscript, but that clearly would delay publication substantially. Another possibility is to mention the new methodology with only brief detail, but to compare the results to a standard existing method for coding syllables. This would give the reader some confidence that, although we do not understand the method the authors used, it is broadly comparable to (but better than!) methods that we do understand. Whatever solution the authors choose, the current presentation is not really acceptable. The somewhat handwaving introduction to the methodology from line 56 onwards does not give the reader confidence. Even with the (excessive) detail of the autoencoder given in the Methods, statements like, “in a manner akin to multidimensional scaling” (line 64) are neither clear enough to the non-expert, nor specific enough for the expert. Similarly, the use of specialist terminology (“stride”, “nonlinearity”) without explanation helps no one. Presenting equations like Equation 2 is also unhelpful unless the context is explained more – and that would push the manuscript to be even more methodological, rather than focussing on the findings.

Following the reviewer’s advice, we now emphasize the information theoretic analysis while discussing our new methodology in only brief detail. We have significantly simplified the automated labeling algorithm, and now use the hand-labeled categories for 3 of 4 songbird species datasets. We agree that this makes the revised manuscript more digestible. We are preparing a second paper about the computational methodology featured in the original version of the paper.

- 3.2. The second problem is that the manuscript doesn’t really present a clear hypothesis, nor explain how the authors’ findings deviate from what might be expected under some (non-presented) null hypothesis. For example, the Mutual information decay fitting (line 310) is a very nice idea, and one that I think probably reveals important results, but the manuscript is lacking any theoretical background to place this in context. Why are you proposing a concatenation of these two particular models? More importantly, how would you expect the MI to decay under different generational models (a null hypothesis)? The authors need to present at least a summary of the MI decay expected under Markovian and various non-Markovian processes, otherwise how can we judge whether the results are unexpected or not? This question is fleetingly addressed on lines 87-90, and genuinely caught my interest! But it’s simply not developed enough, and this really should be expanded, clearer, and the main thrust

of the manuscript, rather than the detail of the neural network. On line 114 the authors state that exponential MI decay indicates a Markov process, but as this is absolutely central to the manuscript, this needs to be explained in much more detail. Indeed, the main result (lines 97-98) and the main hypothesis (lines 120-124) are buried in the text, rather than highlighted. *The revised manuscript now explicitly outlines our competing hypotheses and the expected decay in MI associated with each (lines #103-108).*

- 3.3. Another thing that rankled me right from the outset (although it really shouldn't have done so), was the way that the authors used the terms "distance" and "decay". The abstract (line 12), and the very start of the introduction (line 31) gives no explanation for what this means, and a reader could easily think that the manuscript is about geographical distance between birds! The text moves immediately on to talk about "strength" (line 35), but doesn't say what this means – mutual information is not mentioned until the following sentence. The central idea of "decay in MI" isn't explained until line 72. I think that the manuscript needs to be rewritten, not from the point of view of an information theory expert, but with a more general reader in mind.

We have rewritten several sections in the revised manuscript (beginning in the abstract and continuing throughout the paper) to clarify our use of the terms "distance" and "decay". We now define each term explicitly and standardize the way we use both terms. Where it makes sense, we use the term 'sequential-distance' to denote that we are discussing the distance between elements in a vocal sequence. We provide an explicit description of how sequential distance is computed on lines #89-91. We revised the section that discusses MI, the 'strength' of dependencies between elements, and the decay in MI with distance between elements (lines #91-94), so that the terms are presented logically and in a way that the general reader can understand. The newly added computational models section and the newly added Figure 2 should help the reader understand how different patterns of MI decay relate to the specific hypotheses tested.

- 3.4. I also have one more technical question. On line 106 the authors say that the transition points were close to the bout length. If these bouts are separated by long silences, then they may be effectively different "messages", and so have different statistical properties. Are we therefore surprised by this finding? This should be mentioned and explored.

As detailed in the methods (starting on line #371), the primary analysis examines element distances within a single day, and thus can span bouts separated by varying durations. As we note, the dynamics of these supra-bout element dependencies are governed primarily by a power-law. We settled on the day-wise sampling strategy because a day seemed reasonable as a sampling interval, and also provided us with sufficient data for the analysis. To ensure that this strategy did not induce any adverse results, we now include a range of different control analyses in which we restrict the relevant sampling interval (and thus the length of possible sequence distances) in several ways. We describe these control analyses in the methods (lines #375-393) and reference them in the main text (lines #154-161). In Extended Data Fig. 5 we show that similar patterns of MI decay are observed when the analyses are restricted to only element pairs that occur within the same song bout. Thus, between-bout dependencies do not fully explain our results. In Extended Data Figure 4, we show (panel c & d) that shuffling the bout order within a day does not diminish the improved fits for the concatenative over the exponential model, whereas shuffling the sequence of syllables within a bout destroys all

benefits of the concatenative over the exponential model. In many individual birds, information decay extends beyond the length of a bout, suggesting there is information preserved across bouts and that bouts are not completely discrete. At the same time, in some birds we see essentially no information carried across the length of a bout, meaning that these signals may be effectively discrete messages.

- 3.5. **Line 31: Language must make use of, or empirically does make use of?**

Language empirically does make use of relationships that persist over long sequential distances. We have adjusted the wording of line 27 accordingly, and hope that the next line in the text (line 31 of the original manuscript; line 27 in the revised manuscript) makes this fact clear.

- 3.6. **Lines 38-40: Introduce the hierarchy of generative grammars for those readers who are unfamiliar.**

We appreciate the reviewers' suggestion to introduce the hierarchy of generative grammars to unfamiliar readers. Given the revised focus of the current manuscript, however, we don't think that adding this would improve the paper or help orient the reader to the hypotheses we are trying to test. As noted above, we have revised the section of the introduction that differentiates our approach and use of hierarchy from that pertaining to phrasal syntax (see response 2.3), and we have more clearly laid out our hypotheses in the context of predicted patterns of MI decay given an exponential, a pure power-law, or a combination of the two (see response 1.2). Discussing the classic hierarchy of generative grammars would be tangential to the scope in the current revision of our manuscript. It is not our goal to tie our observations in speech or birdsong to specific grammars within the Chomsky hierarchy. Instead, we want to tie the statistical structure in speech and birdsong to underlying Markovian and hierarchical processes that generate functionally similar dynamics. Of course, these similar dynamics could be generated by different underlying mechanisms. The specific identity of these mechanisms is not the subject of this paper, and we allow for the possibility that they may differ between songbirds and humans and/or between different songbird species.

- 3.7. **Line 67: I would like to see the accuracy of the validation presented here.**

We think this suggestion pertains to the unsupervised dimensionality reduction and clustering techniques used in the original manuscript, and therefore is not relevant to the current revision, as we are no longer using those techniques. We do use segmentation and unsupervised cluster for the starling dataset in the current revision, but this dataset has no ground-truth labeling and thus cannot be validated.

- 3.8. **Line 76: The speech dataset hasn't been mentioned before. Introduce it before referring to it.**
We have added a section to the introduction (paragraph 4 starting on line #68) introducing each dataset in more detail.

- 3.9. **Line 139: I don't know what this means. "If human proto-languages shared similar hierarchical dependencies, this could provide a morphological substrate for human syntax to exploit as the target of evolution rather than the product."**

We removed this sentence along with our discussions about language evolution. The sentence was proposing that the hierarchical (phrasal) syntax of modern language may have emerged from a selection process that targeted other hierarchical, but non-syntactic, mechanisms. We still like this hypothesis, but recognize that it is speculative, and can't be supported (or disproved) from our results. The revised manuscript emphasizes the observed parallels between the sequential structure of speech and birdsong, rather than any evolutionary implications this might (or might not) hold.

- 3.10. *Lines 144-161: This is such a diverse set of sources that you should discuss the implications in the text. Some are wild, some captive (and captive reared?). Although I don't think it necessarily detracts from the conclusions, it's still important to give some discussion. As noted above (see response 3.8), we added a section to the introduction (starting on line #68) that introduces each dataset in detail.*

- 3.11. *Line 191: What is your justification for this? The whole segmentation process is quite novel and a little quirky, so (if this manuscript were focussed on the methodology) it would be good to give some graphs to illustrate.*
We find that the dynamic threshold setting in the segmentation algorithm provides better segmentation in the context of varying background noise that is present in different sound files. As noted above (see response 3.1), we no longer emphasize our novel computational methods in the revised manuscript and focus instead on the results. The segmentation is now only relevant to the Starling dataset. We recognize that our segmentation algorithms are not perfect and that they allow for some noise in the way that different instances of a similar syllable may be segmented. In general, this noise will inflate the number of potential syllable categories and tend to weaken any relationship between elements that are incorrectly assigned to different categories. Thus these errors will work to increase the noise in our MI estimates for the starling dataset. Because the staling results are qualitatively identical to those for the other species, we don't think this potential segmentation noise is problematic. As a partial control, we also parsed the songs of one starling according to fixed time durations (ranging from 0.001 to 1 sec) and classified the waveforms using K-means (Extended Data Figure 8). This analysis, which produced much noisier sound 'categories' than the syllable-level segmentation, yields a pattern of MI decay qualitatively similar to that shown in Figure 4. This suggests that the reported results in the revised manuscript are robust to a wide range of segmentation procedures.

Bibliography

- Altmann, E.G., Cristadoro, G. and Esposti, M.D. 2012. On the origin of long-range correlations in texts. *Proceedings of the National Academy of Sciences of the United States of America* 109(29), pp. 11582–11587.
- Cody, M.L., Stabler, E., Sánchez Castellanos, H.M. and Taylor, C.E. 2016. Structure, syntax and “small-world” organization in the complex songs of California Thrashers (*Toxostoma redivivum*). *Bioacoustics* 25(1), pp. 41–54.

- Ebeling, W. and Neiman, A. 1995. Long-range correlations between letters and sentences in texts. *Physica A: Statistical Mechanics and its Applications* 215(3), pp. 233–241.
- Hyland Bruno, J. and Tchernichovski, O. 2017. Regularities in zebra finch song beyond the repeated motif. *Behavioural Processes*.
- Levitin, D.J., Chordia, P. and Menon, V. 2012. Musical rhythm spectra from Bach to Joplin obey a $1/f$ power law. *Proceedings of the National Academy of Sciences of the United States of America* 109(10), pp. 3716–3720.
- Li, W. and Kaneko, K. 1992. Long-range correlation and partial $1/f\alpha$ spectrum in a noncoding DNA sequence. *EPL (Europhysics Letters)*.
- Lin, H.W. and Tegmark, M. 2016. Critical Behavior from Deep Dynamics: A Hidden Dimension in Natural Language.
- Markowitz, J.E., Ivie, E., Kligler, L. and Gardner, T.J. 2013. Long-range order in canary song. *PLoS Computational Biology* 9(5), p. e1003052.
- Pearre, B., Perkins, L.N., Markowitz, J.E. and Gardner, T.J. 2017. A fast and accurate zebra finch syllable detector. *Plos One* 12(7), p. e0181992.
- Sasahara, K., Cody, M.L., Cohen, D. and Taylor, C.E. 2012. Structural design principles of complex bird songs: a network-based approach. *Plos One* 7(9), p. e44436.
- Suzuki, R., Buck, J.R. and Tyack, P.L. 2006. Information entropy of humpback whale songs. *The Journal of the Acoustical Society of America* 119(3), pp. 1849–1866.

Reviewers' Comments:

Reviewer #1:

Remarks to the Author:

I thank the authors, both for responding so thoroughly to my comments and to those of the other reviewers and for a much improved manuscript. I had a few remaining points to address.

Line 10: This is explained better in the main text, but here in the beginning of the abstract it was unclear whether "long distances" referred to geographic distance or distance between words.

Line 22: "functionally equivalent dynamics" -- unclear if equivalent between birds and humans or between Markovian and hierarchical.

Line 37: "symbols" here was unclear because you had talked about numerous components in the previous sentence. Does it refer to any of the components or only characters?

Line 56: Since you quantify both human and non-human vocal communication, maybe this sentence could highlight that comparison? Particularly because on line 178 the authors say that their results change our understanding of both communication systems.

Line 90-93: I appreciated the definition of MI earlier in the text, and the definition of sequential distance here. Perhaps here the authors could give slightly more intuition for the reader about how MI is computed?

Line 136: "To keep consistent analyses across languages and songbird species we use distances up to 100 syllables in each analysis" → this makes it seem like you are looking at syllables in human languages, but the sentences before and after specify phonemes as the unit of analysis.

Line 159: I came back to this once I read the full description of the shuffling analysis in lines 381-390, and I think the authors could describe the within-bout shuffling here, as well. It is cool that the power-law dynamics exist even within the bout, but it feels like an important omission that these dynamics seem to *depend* on within-bout structure.

Line 160: "a number of songbirds" -- not clear here if you are referring to species or individual songbirds.

Line 184: It might be helpful to have a short summary of refs 18-24 here.

Line 238: I would like slightly more description of the similarities and differences between the language corpora so that I can compare the results across languages. Perhaps Extended Data 2 could have more rows (presence of syllables, mora, parts of speech, etc., maybe an explanation of the differences between syllables and mora) Are they all translated to IPA? Does the writing system (Kana vs. Latin vs. IPA?) have an impact on the analysis here? Was any other processing done to standardize the databases?

Line 262: typo "annotation and algorithm"

Line 336: I would have liked a little more detail on the shuffling process.

Line 381-390: I found the results of this shuffling analysis to be somewhat inconsistent with the results of the main analysis. A main result is that the concatenative model is the best fit for both

birdsong and language. For languages, the authors state that “the exponential component contributes most strongly at short distances between phonemes, at the scale of words, while the power-law primarily governs longer distances between phonemes, presumably reflecting relationships between words,” and for bird songs, “That is, both components contribute at all element distances, but exponential decay is strongest at short syllable distances, and power-law decay primarily governs longer-distance syllable dependencies.” I came away with a different intuition from the shuffling analysis, though. The authors state “We find that when the within-bout structure is removed the concatenative model does not show an improvement over the exponential model, while when the between-bout structure is removed, the concatenative model continues to show an improvement over the exponential model.” Thus, if the authors shuffle the syllables within a song (altering short distances), the power-law portion of the concatenative model goes away, but if they shuffle the order of bouts (altering long distances), the power-law portion of the concatenative model is still a good fit. These two results (from the main analysis and the shuffling analysis) seem to be at odds with one another, since the power-law was stated to govern the long distance dependencies. I would like to see this fleshed out more in the main text.

Line 383: I wanted to know the criteria the authors used to define a bout -- how much more silence separates a bout than separates a syllable?

Figure 1. The caption says “Each row in the figure corresponds to the same animal” -- I think it should say “Each column...”?

Figure 2: Are the parameters of these models available somewhere?

Figure 3: I had a question about the last row, the probability density of phonemes. It seemed surprising that there wasn't more of a dropoff on the left side of each distribution. How many one-phoneme words are in each of these languages?

Figure 4: From panel c of the Bengalese finch (purple), it seems like subtracting the power law leaves more noise in panel c of this species than the other three species (i.e. it seems like the power law has less explanatory power in this species). Do the authors have an intuition about this result? This pattern is not really seen in the other bird species or the languages.

Extended Data Fig 1: “The minimum in curvature for words, part-of-speech, and syllables is shorter than in phonemes” Does this mean shorter as measured in phonemes, or shorter in words, part-of-speech, or syllables, which are made up of multiple phonemes?

Extended Data Fig. 3: “each row corresponds to a single individual” -- I think there are two individuals per row.

Reviewer #2:

Remarks to the Author:

Re-Review of Sainburg ... Gentner

This is a substantially revised ms. which is much improved and has taken the critiques of the initial ms to heart quite thoroughly by adding new a species, new languages and new analyses (and by dropping the unconvincing “language evolution” angle). I think it is much better as a result, and much more suitable for a broad general audience. I now recommend acceptance with only a few minor revisions.

1. The only thing unconvincing in the current ms. is the characterization of the "concatenative" model. Although the formulae on page 18 make clear that it is simple a linear combination of the power law and exponential equations, this insight might not be accessible to all readers. I suggest a better term for this than "concatenative" would be "hybrid" (or bipartite, or mixed, etc). Where the AIC is discussed in the Methods it should be specified how many free parameters are present in each model (3 for the first two and 5 for the last) – again clear in the equations but worth stating verbally. I would also suggest a few sentences on how this model's predictions vary from the Hidden Markov models that play such an extensive role in the speech processing literature, and have occasionally be applied to animal vocalizations as well:

Katahira, K., Suzuki, K., Kagawa, H., and Okanoya, K. (2013). "A simple explanation for the evolution of complex song syntax in Bengalese finches," *Biology Letters* 9, 20130842.

Katahira, K., Suzuki, K., Okanoya, K., and Okada, M. (2011). "Complex Sequencing Rules of Birdsong Can be Explained by Simple Hidden Markov Processes," *PLoS ONE* 6, e24516.

Mellinger, D. K., and Clark, C. W. (2000). "Recognizing transient low-frequency whale sounds by spectrogram correlation," *J. Acoustic. Soc. Am.* 107, 3518-3529.

Reby, D., Andre-Obrecht, R., Galinier, A., Farinas, J., and Cargnelutti, B. (2006). "Cepstral coefficients and hidden Markov models reveal idiosyncratic voice characteristics in red deer (*Cervus elaphus*) stags," *J. Acoustic. Soc. Am.* 120, 4080-4089.

2. Regarding "power laws", watch the hyphens: "power law" when alone (e.g. in abstract) but "power-law distribution" when it's a modifier.

3. The sentence on page 7 is unnecessarily complicated by the "respectively" construction. This should be broken into two sentences, one for each type of decay/dynamics.

4. There are still many (probably bibtex generated) errors of capitalization in the reference list. For example:

10. bach, joplin

35 california, toxostoma

Many journal names e.g. *Animal behavior*, *Computational linguistics*

pg 10 "int some species" delete t

Reviewer #3:

Remarks to the Author:

The revised manuscript is obviously greatly improved, much clearer, and it is now much easier to see the case that the authors are making.

However, I am still concerned about the lack of a null hypothesis. Reading the manuscript carefully, I think that the authors do, in fact, perform the required tests, but they do not report them in detail, and only mention their null models towards the end of the Methods. Hopefully, they will be able to use their existing data and analyses to incorporate such tests into the manuscript without too much difficulty. I do, however, consider this a "major revision", because it is really an essential leg that underpins the entire argument.

The main problem is that the authors test three models (exponential, power, and combined), and find that the combined model fits best (e.g. line 112). This is of course to be expected, and so a null hypothesis test is required. The general argument made on lines 124-139 is well formulated and clearly presented, and could be convincing. If the appropriate tests were performed, including a test randomising at the word and phoneme levels.

This appears to be what the authors discuss on lines 383-390.

“We did this by shuffling the ordering of bouts within a day but retaining the within-bout structure, and by shuffling the within-bout sequences but retaining the ordering of bouts”

However, a detailed explanation of what is done here must be given. Then, this section must be moved right up to the front, or at least referred to at the beginning of the Methods section, to indicate how null models are being compared. We also, of course, need to see the results of these comparisons.

Additional points:

On line 156, I do not believe that the AIC comparison here is valid. AIC can be used to compare models with the same data set, not the same model with different data sets.

The observation on lines 179-181 might well be expected, given that the production mechanism may well be constrained in how successive sounds can be generated. The following observation from line 182 onwards is more novel, but a little speculative. It would be nice to suggest why the authors think this is to be expected.

In the equations, all symbols must be defined, especially (15). The discussion of AIC is best placed in an appendix, or simply referred to in a standard text.

The claim on line 150 that the transition between models is of the order of the length of a song in some species is one that I cannot spot from the figure. This should be clarified!

Reviewer #1:

I thank the authors, both for responding so thoroughly to my comments and to those of the other reviewers and for a much improved manuscript. I had a few remaining points to address.

- 1.1. Line 10: This is explained better in the main text, but here in the beginning of the abstract it was unclear whether “long distances” referred to geographic distance or distance between words.
We edited this line. It now reads “Human language possesses a rich hierarchical structure that allows for meaning to be altered by words spaced far apart within or between sentences” (lines #9-10).
- 1.2. Line 22: “functionally equivalent dynamics” -- unclear if equivalent between birds and humans or between Markovian and hierarchical.
We modified this sentence to read, “Thus, the sequential organization of acoustic elements in two learned vocal communication signals (speech and birdsong) show functionally equivalent dynamics, governed by similar processes.” (line #20-22)
- 1.3. Line 37: “symbols” here was unclear because you had talked about numerous components in the previous sentence. Does it refer to any of the components or only characters?
We replaced symbols with “elements (e.g. words, characters)” here, to be more consistent with the rest of the paragraph and remove any ambiguity.
- 1.4. Line 56: Since you quantify both human and non-human vocal communication, maybe this sentence could highlight that comparison? Particularly because on line 178 the authors say that their results change our understanding of both communication systems.
We revised this sentence to state “The present study examines how Markovian and hierarchical processes combine to govern the sequential structure of birdsong and speech.” (lines #55-57)

- 1.5. Line 90-93: I appreciated the definition of MI earlier in the text, and the definition of sequential distance here. Perhaps here the authors could give slightly more intuition for the reader about how MI is computed?
In the methods section, we have a more intuitive explanation of how MI is calculated (lines #415-434) to supplement the earlier definition in the main text (lines #36-37, #90-96). We added a reference to the methods section in the main text where the reviewer noted (line #91).
- 1.6. Line 136: “To keep consistent analyses across languages and songbird species we use distances up to 100 syllables in each analysis” → this makes it seem like you are looking at syllables in human languages, but the sentences before and after specify phonemes as the unit of analysis.
We edited this sentence to read “To keep consistent analyses across languages and songbird species we use distances up to 100 elements (syllables in birdsong and phones in speech) in each analysis” (lines #462-463).
- 1.7. Line 159: I came back to this once I read the full description of the shuffling analysis in lines 381-390, and I think the authors could describe the within-bout shuffling here, as well. It is cool that the power-law dynamics exist even within the bout, but it feels like an important omission that these dynamics seem to *depend* on within-bout structure.
We added several sentences to the main text to more clearly describe the shuffling analyses, and to discuss the within bout structure observation as well as the new shuffling analyses on speech (discussed in 1.13, lines #175-195, lines #243-281).
- 1.8. Line 160: “a number of songbirds” -- not clear here if you are referring to species or individual songbirds.
We revised this sentence, with the new manuscript specifying, “... a number of individual songbirds ...” (line #207).
- 1.9. Line 184: It might be helpful to have a short summary of refs 18-24 here.
We revised this sentence (lines #222-224) by adding more specific examples of the complex sequential dynamics observed in prior studies.
- 1.10. Line 238: I would like slightly more description of the similarities and differences between the language corpora so that I can compare the results across languages. Perhaps Extended Data 2 could have more rows (presence of syllables, mora, parts of speech, etc., maybe an explanation of the differences between syllables and mora) Are they all translated to IPA? Does the writing system (Kana vs. Latin vs. IPA?) have an impact on the analysis here? Was any other processing done to standardize the databases?
We extended the Speech Corpora section in the Methods to add explicit reference to the transcription methods and phonetic dictionaries used for each corpus (lines #316-348). We also added a sentence noting that because these analyses are performed on datasets transcribed using different methods, we cannot confirm how different forms of transcription might affect our results (lines #348-351). As suggested, Extended Data Table 2 now includes the presence of different

units. We also added a sentence about the distinction between mora and syllables in the methods (lines #342-344). Finally, due to the variability in transcription methods and unit definitions used across datasets, we have reworded the text to use the more neutral term ‘phones’, rather than ‘phonemes’, which we had used to specifically describe segmental units from the Buckeye corpus in the initial version of this manuscript.

1.11. Line 262: typo “annotation and algorithm”

This typo has been fixed.

1.12. Line 336: I would have liked a little more detail on the shuffling process.

We added a sentence (line #430-432) further describing the shuffling process: “This shuffling consists of a permutation of each individual sequence being used in the analysis, which differs depending on the type of analysis (e.g. a bout of song in Extended Data Fig. 4, versus an entire day of song in Fig. 4).”

1.13. Line 381-390: I found the results of this shuffling analysis to be somewhat inconsistent with the results of the main analysis. A main result is that the concatenative model is the best fit for both birdsong and language. For languages, the authors state that “the exponential component contributes most strongly at short distances between phonemes, at the scale of words, while the power-law primarily governs longer distances between phonemes, presumably reflecting relationships between words,” and for bird songs, “That is, both components contribute at all element distances, but exponential decay is strongest at short syllable distances, and power-law decay primarily governs longer-distance syllable dependencies.” I came away with a different intuition from the shuffling analysis, though. The authors state “We find that when the within-bout structure is removed the concatenative model does not show an improvement over the exponential model, while when the between-bout structure is removed, the concatenative model continues to show an improvement over the exponential model.” Thus, if the authors shuffle the syllables within a song (altering short distances), the power-law portion of the concatenative model goes away, but if they shuffle the order of bouts (altering long distances), the power-law portion of the concatenative model is still a good fit. These two results (from the main analysis and the shuffling analysis) seem to be at odds with one another, since the power-law was stated to govern the long-distance dependencies. I would like to see this fleshed out more in the main text.

The reviewer makes a good point regarding the ambiguity of our wording in our description of the various effects. To be clear, it is not our intent to exclusively equate the power law to longer-distance dependencies. As we show in figures 3b & 4b for both speech and birdsong, there is a strong and uniform power law evident across dependencies at all distances. That is, the power law governs both short and long-distance dependencies. The power law is most evident in the long-range dependencies because the exponential component is weakest (or absent) in the long-range dependencies. Relatedly, we note that bout length shouldn't be taken as a rigid cutoff for distinguishing between long- and short-distance dependencies. From Figure 4, one can see that there are many bouts whose lengths encompass long-range dependencies at which the power law primarily shapes MI decay. We have revised the wording in several parts of the manuscript

to clarify these subtle points. With these points in mind, it is not hard to see how shuffling within a song bout would affect both the exponential and the power law dependencies. At the same time, we agree that it is reasonable to ask why the power law does not persist when we shuffle the birdsong datasets within a bout. This is particularly interesting given the new analyses of shuffled speech (lines #134-163), which show a clear power law persisting when phonemes are shuffled within words and utterances (utterances are long phrases of speech defined in the dataset transcripts).

The nature of this difference between birdsong and speech is not entirely clear. One possibility we can't rule out is that it reflects relatively uninteresting differences in the relationship between the level of transcription (bouts vs words) and the dataset structure. For the songbirds, we have relatively few bouts per day, whereas for speech we have many words (and utterances) per transcript. Thus in the birdsong data, there are relatively few supra-bout-length dependencies represented in the unshuffled data, and any contribution from them that remains in the shuffled data is below the MI noise floor. A second (and perhaps more interesting) interpretation is that the hierarchical organization of speech and birdsong differ at a more fundamental level. It may be that hierarchical organization extends across much longer timescales and levels of organization (e.g. phonemes, words, part of speech, phrases, etc.) speech, and the underlying Markovian and hierarchical processes may be more separable than in birdsong. We have included a brief discussion of these possible differences at the end of the revised manuscript (lines #252-258). Fleshing out these detailed differences (or the lack thereof) will require the collection of much larger songbird vocalization data sets and development of novel methods for discretization on timescales larger than syllables but shorter than song bouts. This is beyond the scope of the present study. In the revised ms, we extended our discussion of the birdsong dataset to include the results of the shuffling analysis (lines #175-189), and now explicitly mention that the power-law decay is dependent upon within-bout structure, as opposed to the within-word relationships revealed by the new shuffling analysis added to this revision (lines #189-195).

- 1.14. Line 383: I wanted to know the criteria the authors used to define a bout -- how much more silence separates a bout than separates a syllable?

The threshold for the amount of silence that separates a bout was based upon the distribution of inter-syllable-gaps for each species, except in Bengalese finch where the dataset was already segmented into bouts. The threshold was set at 60 seconds for Cassin's vireo and California thrashers, and at 10 seconds for European starlings (line #356). The distribution of inter-syllable intervals is shown below in Response Letter Figure 1.

Response Letter Figure 1: Inter-syllable interval histogram for each bird species.

- 1.15. Figure 1. The caption says “Each row in the figure corresponds to the same animal” -- I think it should say “Each column...”?
We corrected this mistake.
- 1.16. Figure 2: Are the parameters of these models available somewhere?
The parameters for the two Markov models are available in the cited sources. We now give the parameters for the hierarchical model in the methods section on lines #531-534. In addition, all full model implementations (with parameters) are available in the GitHub repository.
- 1.17. Figure 3: I had a question about the last row, the probability density of phonemes. It seemed surprising that there wasn't more of a dropoff on the left side of each distribution. How many one-phoneme words are in each of these languages?
This value ranges from 6% to 18% depending on the language plotted. We note that the probability density plots in the original figure are smoothed with a Gaussian kernel, however, so the severity of any dropoff may be somewhat occluded. We have updated Figures 3 and 4 to show the empirical distributions of sequence lengths with a smoothed kernel superimposed.
- 1.18. Figure 4: From panel c of the Bengalese finch (purple), it seems like subtracting the power law leaves more noise in panel c of this species than the other three species (i.e. it seems like the power law has less explanatory power in this species). Do the authors have an intuition about this result? This pattern is not really seen in the other bird species or the languages.
We agree that the power-law-only fit is noisier in the Bengalese finch than the other three species. We do not presently have an intuition for why this is the case but consider this observation interesting given the number of research papers exploring sequential organization in Bengalese finches. On lines #278-281 we introduce the idea that there may be important variation between species. It is our preference to leave any stronger claims about Bengalese finches for subsequent studies designed explicitly to test for and investigate such species differences.
- 1.19. Extended Data Fig 1: “The minimum in curvature for words, part-of-speech, and syllables is shorter than in phonemes” Does this mean shorter as measured in phonemes, or shorter in words, part-of-speech, or syllables, which are made up of multiple phonemes?
We mean shorter in their respective individual units. This is not surprising because, as the reviewer mentioned, words, parts of speech, and syllables are generally comprised of multiple phonemes. We corrected the caption of this figure (now Extended Data Fig 2) to read, “The minima in curvature for words, part-of-speech, and syllables are shorter (in their respective units) than in phones,...”
- 1.20. Extended Data Fig. 3: “each row corresponds to a single individual” -- I think there are two individuals per row.
We corrected this error (now Extended Data Fig. 5).

Reviewer #2 (in blue):

This is a substantially revised ms. which is much improved and has taken the critiques of the initial ms to heart quite thoroughly by adding new species, new languages and new analyses (and by dropping the unconvincing “language evolution” angle). I think it is much better as a result, and much more suitable for a broad general audience. I now recommend acceptance with only a few minor revisions.

- 2.1. The only thing unconvincing in the current ms. is the characterization of the “concatenative” model. Although the formulae on page 18 make clear that it is simple a linear combination of the power law and exponential equations, this insight might not be accessible to all readers. I suggest a better term for this than “concatenative” would be “hybrid” (or bipartite, or mixed, etc). Where the AIC is discussed in the Methods it should be specified how many free parameters are present in each model (3 for the first two and 5 for the last) – again clear in the equations but worth stating verbally.

We now refer to the ‘concatenative’ model as the ‘composite’ model. We now note the number of parameters in each model in the main text along with a reference to each equation (lines #101-104), and state explicitly that the composite model is a linear combination of the exponential and power-law models (line #103). In addition, we modified the ‘Model selection’ section to discuss why we use AICc (lines #446-456). We also moved the AIC equations to an appendix section at the suggestion of Reviewer #3 (3.4).

- 2.2. I would also suggest a few sentences on how this model’s predictions vary from the Hidden Markov models that play such an extensive role in the speech processing literature, and have occasionally be applied to animal vocalizations as well: [refs omitted]

We have added a short discussion (lines #232-242) to help the reader understand how our results fit in the context of Hidden Markov Models (HMM) which have been used historically to capture the dynamics of speech (until recent improvements in algorithmic approaches for modeling long-term relationships), and in birdsong. Importantly, our results hold regardless of whether one considers a simple Markov Chain (MC) model or a more complex HMM. In both cases, the underlying system is assumed to be a Markov process, in the HMM the states are not directly observed but are “hidden” behind probability distributions. This allows for more flexibility in the structure of an observed sequence and can work better than a MC for modeling sequences where the elements vary from instance to instance but are members of a category, like the same word pronounced slightly differently, hand-written exemplars of digits, or birdsong motifs or syllables. For both the HMM and the MC, transition probabilities between states are defined by a Markov process, and thus the MI between explicit states (in MC) or category tokens (in HMM) will decay exponentially as a function of the distance between states or tokens. Throughout the paper, we have used the more generic term “Markov model”, because it applies to both the simple MC and the HMM.

- 2.3. Regarding “power laws”, watch the hyphens: “power law” when alone (e.g. in abstract) but “power-law distribution” when it’s a modifier.

We have fixed hyphenation throughout the text.

- 2.4. The sentence on page 7 is unnecessarily complicated by the “respectively” construction. This should be broken into two sentences, one for each type of decay/dynamics.

We broke our original sentence into two (lines #130-133): “The observed exponential decay at inter-word distances agrees with the longstanding consensus that phonological organization is governed by regular (or subregular) grammars with Markovian dynamics. The emphasis of a power-law decay at intra-word distances, likewise, agrees with the prior observations of hierarchical long-range organization in language.”

- 2.5. There are still many (probably bibtex generated) errors of capitalization in the reference list. For example: [examples omitted]

We went through the references and fixed capitalization for each reference.

- 2.6. pg 10 “int some species” delete t

We addressed this error.

Reviewer #3 (in blue):

- 3.1. The revised manuscript is obviously greatly improved, much clearer, and it is now much easier to see the case that the authors are making. However, I am still concerned about the lack of a null hypothesis. Reading the manuscript carefully, I think that the authors do, in fact, perform the required tests, but they do not report them in detail, and only mention their null models towards the end of the Methods. Hopefully, they will be able to use their existing data and analyses to incorporate such tests into the manuscript without too much difficulty. I do, however, consider this a “major revision”, because it is really an essential leg that underpins the entire argument. The main problem is that the authors test three models (exponential, power, and combined), and find that the combined model fits best (e.g. line 112). This is of course to be expected, and so a null hypothesis test is required. The general argument made on lines 124-139 is well formulated and clearly presented, and could be convincing. If the appropriate tests were performed, including a test randomising at the word and phoneme levels. This appears to be what the authors discuss on lines 383-390. “We did this by shuffling the ordering of bouts within a day but retaining the within-bout structure, and by shuffling the within-bout sequences but retaining the ordering of bouts” However, a detailed explanation of what is done here must be given. Then, this section must be moved right up to the front, or at least referred to at the beginning of the Methods section, to indicate how null models are being compared. We also, of course, need to see the results of these comparisons.

NB: We asked for clarification, through the editor, on the reasoning behind the reviewer’s expectation that the combined model should fit better than the other models. The reviewer confirmed that this was based on the fact that the combined model has more parameters than the other models.

As the reviewer notes, we tested three models (exponential, power, and combined (now called composite)) and find that the composite model fits the speech and birdsong datasets best. When comparing the fit between different models for a given dataset, particularly when models have different numbers of free parameters as ours do, one must be careful to avoid overfitting. For this reason, we used the AIC estimator of relative model quality as the basis for model selection. The AIC is an established means for model selection that takes into account both the goodness-of-fit and model simplicity, by penalizing increasing numbers of parameters. Because our datasets vary in size, we use the AICc form of the estimator which imposes an additional penalty (beyond the penalty imposed by AIC) that is tied to both the sample size and the second order information loss due to the number of model parameters. Using BIC as a model selection criterion (which imposes a different penalty for increasing parameters) yields identical results.

While we agree with the reviewer that the shuffle analyses can be seen as a control for over-fitting of the composite model, our model selection criteria are well-founded and already guard against this concern.

Regardless of the foregoing justification, we recognize that any confusion or concerns regarding our model selection methods need to be avoided. Accordingly, we have revised the manuscript in several places. We now explicitly note that we are using AICc (and why) when we first describe the model fits. (lines #104-107). We also state that the relative model probabilities are derived from the AICc values given in extended data tables 3 and 4 (line #105, lines #453-456). As suggested, we added additional analyses for the speech data sets where we shuffle within words, between words, within utterances, and between utterances (Extended Data Fig. 1). As suggested, we also now give a detailed description of the shuffling procedures for speech (see 1.12, lines #134-163) and birdsong (lines #175-195) and have moved these results to an earlier position in the manuscript. We added additional detail regarding the shuffles to the revised methods (lines #481-494).

- 3.2. **On line 156, I do not believe that the AIC comparison here is valid. AIC can be used to compare models with the same data set, not the same model with different data sets.**

We agree that AIC cannot be used to compare the same model with different datasets, but that is not how we are using it here. We use the $\Delta AICc$ to compare the exponential vs composite models within the same dataset. We then use a simple correlation to relate the magnitude of the $\Delta AICc$ (within a dataset) to the size of that dataset. We do this to make the point that a smaller dataset tends to show smaller $\Delta AICc$ (see Extended Data Fig. 5). The $\Delta AICc$ is a measure of information and can be used to directly compute the relative likelihood of a given model for a given dataset (Burnham et al. 2011). Comparing these relative likelihoods across datasets shows that with a smaller dataset one has an increased chance selecting the exponential model.

- 3.3. **The observation on lines 179-181 might well be expected, given that the production mechanism may well be constrained in how successive sounds can be generated. The following observation from line 182 onwards is more novel, but a little speculative. It would be nice to suggest why the authors think this is to be expected.**

We have added a short description of the complex dynamics already observed in birdsong, which we propose are suggestive of dynamics more complex than short-range Markovian relationships dominating the signal (lines #49-53, #122-224). See also response 1.9.

- 3.4. *In the equations, all symbols must be defined, especially (15). The discussion of AIC is best placed in an appendix, or simply referred to in a standard text.*

We added definitions to all symbols where they did not already exist. As suggested, we moved the equations for AIC to an appendix.

- 3.5. *The claim on line 150 that the transition between models is of the order of the length of a song in some species is one that I cannot spot from the figure. This should be clarified!*

We agree that the relationship between bout-length and the transition from exponential to power-law dominance in the birdsong signal is not very obvious, especially in light of the new thrasher dataset. We removed this sentence.

---end rebuttal---

Bibliography

Burnham, K.P., Anderson, D.R. and Huyvaert, K.P. 2011. AIC model selection and multimodel inference in behavioral ecology: some background, observations, and comparisons. *Behav Ecol Sociobiol* 65(1), pp. 23–35.

Reviewers' Comments:

Reviewer #1:

Remarks to the Author:

I thank the authors for an improved manuscript! My concerns have been addressed.

A few minor points:

I appreciated Response Letter Figure 1, and I thought it might be nice to include this as a supplemental figure. Particularly for the first two panels, I would like to see the axis extended to 60 seconds.

Regarding the new text on HMM -- I am glad that the authors include this explanation. In the response, the authors state "our results hold regardless of whether one considers a simple Markov chain model or a more complex HMM" -- just to clarify, only the simple Markov model is reported in the text, correct?

Is "phones" defined for language?

I liked the use of "utterances" as a shuffling-strategy. Is it possible to include the distribution of utterance lengths for English and Japanese?

Line 185 - when the authors say "Similar to the shuffling analysis", does this mean the shuffling analysis conducted in languages?

Line 204 - it appears the authors have left in a note to themselves.

Methods: Is it possible to make the startling dataset publicly available? It would be good for repeatability since the code is available, and the paper has benefited from the availability of datasets for other species.

Extended data figure 6: I see the positive correlation with dataset length, but there were still a lot of points right around zero even at large dataset lengths. I think this is related to the authors' statement that the composite model does not fit some individuals; is this the correct interpretation? I'm curious whether there was any readily apparent qualitative difference in these individuals -- repertoire size, etc.

Extended data figure 9: what do the authors think is happening at about 80% of the way across the x-axis for the thrasher?

Reviewer #3:

None

Reviewer #1 (Remarks to the Author):

I thank the authors for an improved manuscript! My concerns have been addressed.

A few minor points:

1.

- 1.1. I appreciated Response Letter Figure 1, and I thought it might be nice to include this as a supplemental figure. Particularly for the first two panels, I would like to see the axis extended to 60 seconds.

We added this figure (inter-syllable interval length across songbird species) with a 60 second x-axis as Supplementary Figure 9, and referenced the figure on line #427.

- 1.2. Regarding the new text on HMM -- I am glad that the authors include this explanation. In the response, the authors state "our results hold regardless of whether one considers a simple Markov chain model or a more complex HMM" -- just to clarify, only the simple Markov model is reported in the text, correct?

Yes, the simple Markov model is reported in the text. An elaboration on the information decay of sequences modeled by HMMs and their relationship to first-order Markov models can be found in Lin and Tegmark (2016).

- 1.3. Is "phones" defined for language?

Phones are used as an umbrella term for the multiple phonetic units in the different speech datasets. We describe each of those units individually in the Speech Corpora section of the methods (lines #329-364).

- 1.4. I liked the use of "utterances" as a shuffling-strategy. Is it possible to include the distribution of utterance lengths for English and Japanese?

We added this figure as Supplementary Figure 1 and referenced the figure on line #146.

- 1.5. Line 185 - when the authors say "Similar to the shuffling analysis", does this mean the shuffling analysis conducted in languages?

This means similar to the shuffling analysis that was conducted in songbirds (line #194). To make this clearer, we now say, "bout shuffling analysis".

- 1.6. Line 204 - it appears the authors have left in a note to themselves.

Thank you for pointing out this error, it has been corrected.

- 1.7. Methods: Is it possible to make the starling dataset publicly available? It would be good for repeatability since the code is available, and the paper has benefited from the availability of datasets for other species.

We uploaded the starling song dataset to a Zenodo repository referenced in the Data Availability statement.

- 1.8. Extended data figure 6: I see the positive correlation with dataset length, but there were still a lot of points right around zero even at large dataset lengths. I think this is related to the authors' statement that the composite model does not fit some individuals; is this the correct interpretation? I'm curious whether there was any readily apparent qualitative difference in these individuals -- repertoire size, etc.

Yes, that is the correct interpretation (e.g. points below the dashed line on Extended Data Figure 6a (now Supplementary Figure 7a)). We did not uncover any clear qualitative differences, but factors underlying this variance between individuals are an interesting question that we would like to continue to pursue.

- 1.9. Extended data figure 9: what do the authors think is happening at about 80% of the way across the x-axis for the thrasher?

Each separate data point in Extended Data Figure 9 (now Supplementary Figure 11) corresponds to a different model. The change in results at that point is likely due to the model fitting parameters slightly differing due to some noise/variance in decay, that yields a slightly higher r^2 when decay is fit up to ~60 syllables than ~50 or ~70 syllables.

References

Lin, H. W., & Tegmark, M. (2016). *Criticality in formal languages and statistical physics*. arXiv preprint arXiv:1606.06737.